# Interventional Few-Shot Learning

**Zhongqi Yue**[1,3], **Hanwang Zhang**[1], **Qianru Sun**[2], **Xian-Sheng Hua**[3]

[1]Nanyang Technological University, [2]Singapore Management University, [3]Alibaba Group
yuez0003@ntu.edu.sg, hanwangzhang@ntu.edu.sg,
qianrusun@smu.edu.sg, xiansheng.hxs@alibaba-inc.com

## Abstract

We uncover an ever-overlooked deficiency in the prevailing Few-Shot Learning (FSL) methods: the pre-trained knowledge is indeed a confounder that limits the performance. This finding is rooted from our causal assumption: a Structural Causal Model (SCM) for the causalities among the pre-trained knowledge, sample features, and labels. Thanks to it, we propose a novel FSL paradigm: Interventional Few-Shot Learning (IFSL). Specifically, we develop three effective IFSL algorithmic implementations based on the backdoor adjustment, which is essentially a causal intervention towards the SCM of many-shot learning: the upper-bound of FSL in a causal view. It is worth noting that the contribution of IFSL is orthogonal to existing fine-tuning and meta-learning based FSL methods, hence IFSL can improve all of them, achieving a new 1-/5-shot state-of-the-art on *mini*ImageNet, *tiered*ImageNet, and cross-domain CUB. Code is released at `https://github.com/yue-zhongqi/ifsl`.

## 1 Introduction

Few-Shot Learning (FSL) — the task of training a model using very few samples — is nothing short of a panacea for any scenario that requires fast model adaptation to new tasks [64], such as minimizing the need for expensive trials in reinforcement learning [29] and saving computation resource for light-weight neural networks [26, 24]. Although we knew that, more than a decade ago, the crux of FSL is to imitate the human ability of transferring prior knowledge to new tasks [17], not until the recent advances in pre-training techniques, had we yet reached a consensus on "what & how to transfer": a powerful neural network $\Omega$ pre-trained on a large dataset $\mathcal{D}$. In fact, the prior knowledge learned from pre-training prospers today's deep learning era, *e.g.*, $\mathcal{D}$ = ImageNet, $\Omega$ = ResNet in visual recognition [23, 22]; $\mathcal{D}$ = Wikipedia, $\Omega$ = BERT in natural language processing [61, 15].

In the context of pre-trained knowledge, we denote the original FSL training set as *support* set $\mathcal{S}$ and the test set as *query* set $\mathcal{Q}$, where the classes in $(\mathcal{S}, \mathcal{Q})$ are unseen (or new) in $\mathcal{D}$. Then, we can use $\Omega$ as a backbone (fixed or partially trainable) for extracting sample representations $\mathbf{x}$, and thus FSL can be achieved simply by *fine-tuning* the target model on $\mathcal{S}$ and test it on $\mathcal{Q}$ [11, 16]. However, the fine-tuning only exploits the $\mathcal{D}$'s knowledge on "what to transfer", but neglects "how to transfer". Fortunately, the latter can be addressed by applying a post-pre-training and pre-fine-tuning strategy: *meta-learning* [52]. Different from fine-tuning whose goal is the "model" trained on $\mathcal{S}$ and tested on $\mathcal{Q}$, meta-learning aims to learn the "meta-model"

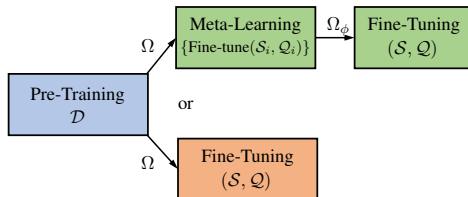

Figure 1: The relationships among different FSL paradigms (color green and orange). Our goal is to remove the deficiency introduced by Pre-Training.

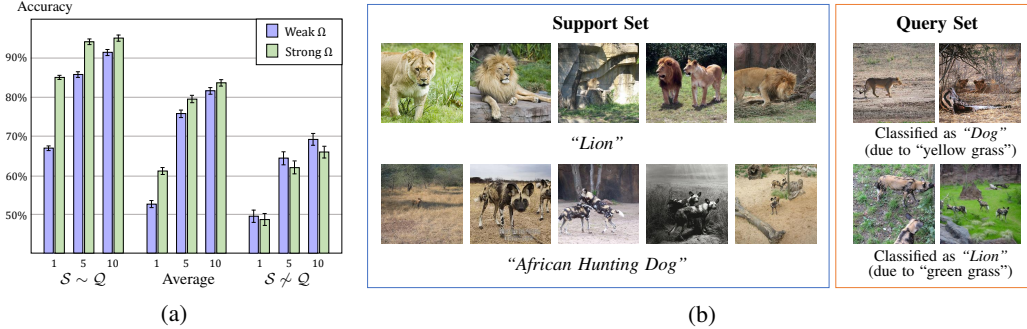

Figure 2: Quantitative and qualitative evidences of pre-trained knowledge misleading the fine-tune FSL paradigm. (a) *mini*ImageNet fine-tuning accuracy on 1-/5-/10-shot FSL using weak and strong backbones: ResNet-10 and WRN-28-10. $\mathcal{S} \sim \mathcal{Q}$ (or $\mathcal{S} \not\sim \mathcal{Q}$) denotes the pre-trained classifier scores of the query is similar (or dissimilar) to that of the support set. "Average" is the mean of both. The dissimilarity is measured using query hardness defined in Section 5.1. (b) An example of 5-shot $\mathcal{S} \not\sim \mathcal{Q}$.

— a learning behavior — trained on many learning episodes $\{(\mathcal{S}_i, \mathcal{Q}_i)\}$ sampled from $\mathcal{D}$ and tested on the target task $(\mathcal{S}, \mathcal{Q})$. In particular, the behavior can be parametrized by $\phi$ using model parameter generator [46, 19] or initialization [18]. After meta-learning, we denote $\Omega_\phi$ as the new model starting point for the subsequent fine-tuning on target task $(\mathcal{S}, \mathcal{Q})$. Figure 1 illustrates the relationships among the above discussed FSL paradigms.

It is arguably a common sense that the stronger the pre-trained $\Omega$ is, the better the downstream model will be. However, we surprisingly find that this may not be always the case in FSL. As shown in Figure 2(a), we can see a paradox: though stronger $\Omega$ improves the performance on average, it indeed degrades that of samples in $\mathcal{Q}$ dissimilar to $\mathcal{S}$. To illustrate the "dissimilar", we show a 5-shot learning example in Figure 2(b), where the prior knowledge on "green grass" and "yellow grass" is misleading. For example, the "Lion" samples in $\mathcal{Q}$ have "yellow grass", hence they are misclassified as "Dog" whose $\mathcal{S}$ has major "yellow grass". If we use stronger $\Omega$, the seen old knowledge ("grass" & "color") will be more robust than the unseen new knowledge ("Lion" & "Dog"), and thus the old becomes even more misleading. We believe that such a paradox reveals an unknown systematic deficiency in FSL, which has been however hidden for years by our gold-standard "fair" accuracy, averaged over all the random $(\mathcal{S}, \mathcal{Q})$ test trials, regardless of the similarity between $\mathcal{S}$ and $\mathcal{Q}$ (*cf.* Figure 2(a)). Though Figure 2 only illustrates the fine-tune FSL paradigm, the deficiency is expected in the meta-learning paradigm, as fine-tune is also used in each meta-train episode (Figure 1). We will analyze them thoroughly in Section 5.

In this paper, we first point out that the cause of the deficiency: pre-training can do evil in FSL, and then propose a novel FSL paradigm: Interventional Few-Shot Learning (IFSL), to counter the evil. Our theory is based on the assumption of the *causalities* among the pre-trained knowledge, few-shot samples, and class labels. Specifically, our contributions are summarized as follows.

- We begin with a Structural Causal Model (SCM) assumption in Section 2.2, which shows that the pre-trained knowledge is essentially a *confounder* that causes spurious correlations between the sample features and class labels in support set. As an intuitive example in Figure 2(b), even though the "grass" feature is not the cause of the "Lion" label, the prior knowledge on "grass" still confounds the classifier to learn a correlation between them.

- In Section 2.3, we illustrate a causal justification of why the proposed IFSL fundamentally works better: it is essentially a causal approximation to many-shot learning. This motivates us to develop three effective implementations of IFSL using the backdoor adjustment [44] in Section 3.

- Thanks to the causal intervention, IFSL is naturally orthogonal to the downstream fine-tuning and meta-learning based FSL methods [18, 62, 27]. In Section 5.2, IFSL improves all baselines by a considerable margin, achieving new 1-/5-shot state-of-the-arts: 73.51%/83.21% on *mini*ImageNet [62], 83.07%/88.69% on *tiered*ImageNet [49], and 50.71%/64.43% on cross-domain CUB [65].

- We further diagnose the detailed performances of FSL methods across different similarities between $\mathcal{S}$ and $\mathcal{Q}$. We find that IFSL outperforms all baselines in every inch.

# 2 Problem Formulations

## 2.1 Few-Shot Learning

We are interested in a prototypical FSL: train a $K$-way classifier on an $N$-shot support set $\mathcal{S}$, where $N$ is a small number of training samples per class (*e.g.*, $N$=1 or 5); then test the classifier on a query set $\mathcal{Q}$. As illustrated in Figure 1, we have the following two paradigms to train the classifier $P(y|\mathbf{x};\theta)$, predicting the class $y \in \{1,...,K\}$ of a sample $\mathbf{x}$:

**Fine-Tuning**. We consider the prior knowledge as the sample feature representation $\mathbf{x}$, encoded by the pre-trained network $\Omega$ on dataset $\mathcal{D}$. In particular, we refer $\mathbf{x}$ to the output of the frozen sub-part of $\Omega$ and the rest trainable sub-part of $\Omega$ (if any) can be absorbed into $\theta$. We train the classifier $P(y|\mathbf{x};\theta)$ on the support set $\mathcal{S}$, and then evaluate it on the query set $\mathcal{Q}$ in a standard supervised way.

**Meta-Learning**. Yet, $\Omega$ only carries prior knowledge in a way of "representation". If the dataset $\mathcal{D}$ can be re-organized as the training episodes $\{(\mathcal{S}_i, \mathcal{Q}_i)\}$, each of which can be treated as a "sandbox" that has the same $N$-shot-$K$-way setting as the target $(\mathcal{S}, \mathcal{Q})$. Then, we can model the "learning behavior" from $\mathcal{D}$ parameterized as $\phi$, which can be learned by the above fine-tuning paradigm for each $(\mathcal{S}_i, \mathcal{Q}_i)$. Formally, we denote $P_\phi(y|\mathbf{x};\theta)$ as the enhanced classifier equipped with the learned behavior. For example, $\phi$ can be the classifier weight generator [19], distance kernel function in $k$-NN [62], or even $\theta$'s initialization [18]. Considering $L_\phi(\mathcal{S}_i, \mathcal{Q}_i; \theta)$ as the loss function of $P_\phi(y|\mathbf{x};\theta)$ trained on $\mathcal{S}_i$ and tested on $\mathcal{Q}_i$, we can have $\phi \leftarrow \arg\min_{(\phi,\theta)} \mathbb{E}_i [L_\phi(\mathcal{S}_i, \mathcal{Q}_i; \theta)]$, and then we fix the optimized $\phi$ and fine-tune for $\theta$ on $\mathcal{S}$ and test on $\mathcal{Q}$. Please refer to Appendix 5 for the details of various fine-tuning and meta-learning settings.

## 2.2 Structural Causal Model

From the above discussion, we can see that $(\phi, \theta)$ in meta-learning and $\theta$ in fine-tuning are both dependent on the pre-training. Such "dependency" can be formalized with a Structural Causal Model (SCM) [44] proposed in Figure 3(a), where the nodes denote the abstract data variables and the directed edges denote the (functional) causality, *e.g.*, $X \rightarrow Y$ denotes that $X$ is the cause and $Y$ is the effect. Now we introduce the graph and the rationale behind its construction at a high-level. Please see Section 3 for the detailed functional implementations.

$\boldsymbol{D \rightarrow X}$. We denote $X$ as the feature representation and $D$ as the pre-trained knowledge, *e.g.*, the dataset $\mathcal{D}$ and its induced model $\Omega$. This link assumes that the feature $X$ is extracted by using $\Omega$.

$\boldsymbol{D \rightarrow C \leftarrow X}$. We denote $C$ as the transformed representation of $X$ in the low-dimensional manifold, whose base is inherited from $D$. This assumption can be rationalized as follows. 1) $D \rightarrow C$: a set of data points are usually embedded in a low-dimensional manifold. This finding can be dated back to the long history of dimensionality reduction [59, 50]. Nowadays, there are theoretical [3, 8] and empirical [77, 71] evidences

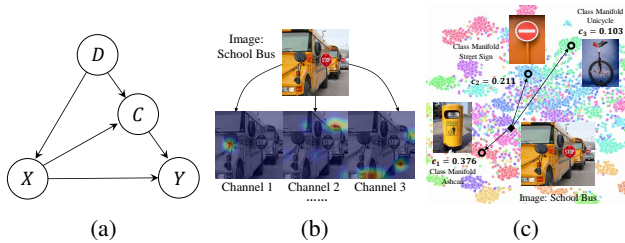

(a)       (b)       (c)

Figure 3: (a) Causal Graph for FSL; (b) Feature-wise illustration of $D \rightarrow C$: Feature channels of pre-trained network(*e.g.* $1 \ldots 512$ for ResNet-10). $X \rightarrow C$: Per-channel response to an image ("school bus") visualized by CAM[77]; (c) Class-wise illustration for $D \rightarrow C$: features are clustered according to the pre-training semantic classes (colored t-SNE plot[37]). $X \rightarrow C$: An image ("school bus") can be represented in terms of the similarities among the base classes ("ashcan", "unicycle", "sign").

showing that disentangled semantic manifolds emerge during training deep networks. 2) $X \rightarrow C$: features can be represented using (or projected onto) the manifold base linearly [60, 9] or non-linearly [6]. In particular, as later discussed in Section 3, we explicitly implement the base as feature dimensions (Figure 3(b)) and class-specific mean features (Figure 3(c)).

$\boldsymbol{X \rightarrow Y \leftarrow C}$. We denote $Y$ as the classification effect (*e.g.*, logits), which is determined by $X$ via two ways: 1) the direct $X \rightarrow Y$ and 2) the mediation $X \rightarrow C \rightarrow Y$. In particular, the first way can be removed if $X$ can be fully represented by $C$ (*e.g.*, feature-wise adjustment in Section 3). The second way is inevitable even if the classifier does not take $C$ as an explicit input, because any $X$ can be

inherently represented by $C$. To illustrate, suppose that $X$ is a linear combination of two base vectors plus a noise residual: $\mathbf{x} = c_1\mathbf{b}_1 + c_2\mathbf{b}_2 + \mathbf{e}$, any classifier $f(\mathbf{x}) = f(c_1\mathbf{b}_1 + c_2\mathbf{b}_2 + \mathbf{e})$ will implicitly exploit the $C$ representation in terms of $\mathbf{b}_1$ and $\mathbf{b}_2$. In fact, this assumption also fundamentally validates unsupervised representation learning [5]. To see this, if $C \nrightarrow Y$ in Figure 3(a), uncovering the latent knowledge representation from $P(Y|X)$ would be impossible, because the only path left that transfers knowledge from $D$ to $Y$: $D \to X \to Y$, is cut off by conditioning on $X$: $D \nrightarrow X \to Y$.

An ideal FSL model should capture the true causality between $X$ and $Y$ to generalize to unseen samples. For example, as illustrated in Figure 2(b), we expect that the "Lion" prediction is caused by the "lion" feature *per se*, but not the background "grass". However, from the SCM in Figure 3(a), the conventional correlation $P(Y|X)$ fails to do so, because the increased likelihood of $Y$ given $X$ is not only due to "X causes Y" via $X \to Y$ and $X \to C \to Y$, but also the spurious correlation via 1) $D \to X$, *e.g.*, the "grass" knowledge generates the "grass" feature, and 2) $D \to C \to Y$, *e.g.*, the "grass" knowledge generates the "grass" semantic, which provides useful context for "Lion" label. Therefore, to pursue the true causality between $X$ and $Y$, we need to use the *causal intervention* $P(Y|do(X))$ [45] instead of the likelihood $P(Y|X)$ for the FSL objective.

## 2.3   Causal Intervention via Backdoor Adjustment

By now, an astute reader may notice that the causal graph in Figure 3(a) is also valid for Many-Shot Learning (MSL), *i.e.*, conventional learning based on pre-training. Compared to FSL, the $P(Y|X)$ estimation of MSL is much more robust. For example, on *mini*ImageNet, a 5-way-550-shot fine-tuned classifier can achieve 95% accuracy, while a 5-way-5-shot one only obtains 79%. We used to blame FSL for insufficient data by the law of large numbers in point estimation [14]. However, it does not answer why MSL converges to the true causal effects as the number of samples increases infinitely. In other words, why $P(Y|do(X)) \approx P(Y|X)$ in MSL while $P(Y|do(X)) \napprox P(Y|X)$ in FSL?

To answer the question, we need to incorporate the endogenous feature sampling $\mathbf{x} \sim P(X|I)$ into the estimation of $P(Y|X)$, where $I$ denotes the sample ID. We have $P(Y|X = \mathbf{x}_i) \coloneqq \mathbb{E}_{\mathbf{x} \sim P(X|I)} P(Y|X = \mathbf{x}, I = i) = P(Y|I)$, *i.e.*, we can use $P(Y|I)$ to estimate $P(Y|X)$. In Figure 4(a), the causal relation between $I$ and $X$ is purely $I \to X$, *i.e.*, $X \to I$ does not exist, because tracing the $X$'s ID out of many-shot samples is like to find a needle in a haystack, given the nature that a DNN feature is an abstract and diversity-reduced representation of many samples [21]. However, as shown in Figure 4(b), $X \to I$ persists in FSL, because it is much easier for a model to "guess" the correspondence, *e.g.*, the 1-shot extreme case that has a trivial 1-to-1 mapping for $X \leftrightarrow I$. Therefore, as we formally show in Appendix 1, the key causal difference between MSL and FSL is: MSL essentially makes $I$ an *instrumental variable* [1] that achieves $P(Y|X) \coloneqq P(Y|I) \approx P(Y|do(X))$. Intuitively, we can see that all the causalities between $I$ and $D$ in MSL are all blocked by colliders[1], making $I$ and $D$ independent. So, the feature $X$ is essentially "intervened" by $I$, no longer dictated by $D$, *e.g.*, neither "yellow grass" nor "green grass" dominates "Lion" in Figure 2(b), mimicking the casual intervention by controlling the use of pre-trained knowledge.

In this paper, we propose to use the backdoor adjustment [44] to achieve $P(Y|do(X))$ without the need for many-shot, which certainly undermines the definition of FSL. The backdoor adjustment assumes that we can observe and stratify the confounder, *i.e.*, $D = \{d\}$, where each $d$ is a stratification of the pre-trained knowledge. Formally, as shown in Appendix 2, the backdoor adjustment for the graph in Figure 3(a) is:

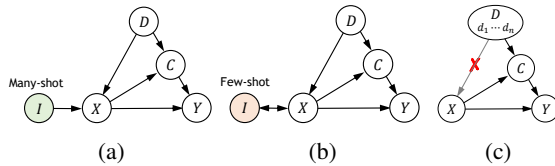

(a)                    (b)                    (c)

Figure 4: Causal graphs with sampling process. (a) Many-Shot Learning, where $P(Y|X) \approx P(Y|do(X))$; (b) Few-Shot Learning where $P(Y|X) \napprox P(Y|do(X))$; (c) Interventional Few-Shot Learning where we directly model $P(Y|do(X))$.

$$P(Y|do(X = \boldsymbol{x})) = \sum_d P(Y|X = \boldsymbol{x}, D = d, C = g(\mathbf{x}, d)) \, P(D = d), \qquad (1)$$

where $g$ is a function defined later. However, it is not trivial to instantiate $d$, especially when $D$ is a 3rd-party delivered pre-trained network where the dataset is unobserved [20]. Next, we will offer three practical implementations of Eq. (1) for Interventional FSL.

## 3   Interventional Few-Shot Learning

Our implementation idea is inspired from the two inherent properties of any pre-trained DNN. First, each feature dimension carries a semantic meaning, *e.g.*, every channel in convolutional neural network is well-known to encode visual concepts [77, 71]. So, each feature dimension represents a piece of knowledge. Second, most prevailing pre-trained models use a classification task as the objective, such as the 1,000-way classifier of ResNet [23] and the token predictor of BERT [15]. Therefore, the classifier can be considered as the distilled knowledge, which has been already widely adopted in literature [24]. Next, we will detail the proposed Interventional FSL (IFSL) by providing three *different* implementations[2] for $g(\mathbf{x}, d)$, $P(Y|X, D, C)$, and $P(D)$ in Eq. (1). In particular, the exact forms of $P(Y|\cdot)$ across different classifiers are given in Appendix 5.

**Feature-wise Adjustment**. Suppose that $\mathcal{F}$ is the index set of the feature dimensions of $\mathbf{x}$, *e.g.*, from the last-layer of the pre-trained network $\Omega$. We divide $\mathcal{F}$ into $n$ equal-size disjoint subsets, *e.g.*, the output feature dimension of ResNet-10 is 512, if $n = 8$, the $i$-th set will be a feature dimension index set of size 512/8 = 64, *i.e.*, $\mathcal{F}_i = \{64(i-1)+1, ..., 64i\}$. The stratum set of pre-trained knowledge is defined as $D := \{d_1, \ldots, d_n\}$, where each $d_i = \mathcal{F}_i$.

**(i)** $g(\mathbf{x}, d_i) := \{k|k \in \mathcal{F}_i \cap \mathcal{I}_t\}$, where $\mathcal{I}_t$ is an index set whose corresponding absolute values in $\mathbf{x}$ are larger than the threshold $t$. The reason is simple: if a feature dimension is inactive in $\mathbf{x}$, its corresponding adjustment can be omitted. We set $t$=1e-3 in this paper.

**(ii)** $P(Y|X, D, C) = P(Y|[\mathbf{x}]_c)$, where $c = g(\mathbf{x}, d_i)$ is implemented as the index set defined above, $[\mathbf{x}]_c = \{x_k\}_{k \in c}$ is a feature selector which selects the dimensions of $\mathbf{x}$ according to the index set $c$. The classifier takes the adjusted feature $[\mathbf{x}]_c$ as input. Note that $d$ is already absorbed in $c$, so $[\mathbf{x}]_c$ is essentially a function of $(X, D, C)$.

**(iii)** $P(d_i) = 1/n$, where we assume a uniform prior for the adjusted features.

**(iv)** The overall feature-wise adjustment is:

$$P(Y|do(X = \boldsymbol{x})) = \frac{1}{n}\sum_{i=1}^{n} P(Y|[\mathbf{x}]_c), \quad \text{where } c = \{k|k \in \mathcal{F}_i \cap \mathcal{I}_t\}. \tag{2}$$

It is worth noting that the feature-wise adjustment is always applicable, as we can always have the feature representation $\mathbf{x}$ from the pre-trained network. Interestingly, our feature-wise adjustment sheds some light on the theoretical justifications for the multi-head trick in transformers [61]. We will explore this in future work.

**Class-wise Adjustment**. Suppose that there are $m$ pre-training classes, denoted as $\mathcal{A} = \{a_1, \ldots a_m\}$. In class-wise adjustment, each stratum of pre-trained knowledge is defined as a pre-training class, *i.e.*, $D := \{d_1, \ldots, d_m\}$ and each $d_i = a_i$.

**(i)** $g(\mathbf{x}, d_i) := P(a_i|\mathbf{x})\bar{\mathbf{x}}_i$, where $P(a_i|\mathbf{x})$ is the pre-trained classifier's probability output that $\mathbf{x}$ belongs to class $a_i$, and $\bar{\mathbf{x}}_i$ is the mean feature of pre-training samples from class $a_i$. Note that unlike feature-wise adjustment where $c$ is an index set, here $c = g(\mathbf{x}, d_i)$ is implemented as a real vector.

**(ii)** $P(Y|X, D, C) = P(Y|\mathbf{x} \oplus g(\mathbf{x}, d_i))$, where $\oplus$ denotes vector concatenation.

**(iii)** $P(d_i) = 1/m$, where we assume a uniform prior of each class.

**(iv)** The overall class-wise adjustment is:

$$P(Y|do(X = \mathbf{x})) = \frac{1}{m}\sum_{i=1}^{m} P(Y|\mathbf{x} \oplus P(a_i|\mathbf{x})\bar{\mathbf{x}}_i) \approx P(Y|\mathbf{x} \oplus \frac{1}{m}\sum_{i=1}^{m} P(a_i|\mathbf{x})\bar{\mathbf{x}}_i), \tag{3}$$

where we adopt the Normalized Weighted Geometric Mean (NWGM) [66, 67] approximation to move the outer sum $\sum P$ into the inner $P(\sum)$. This greatly reduces the network forward-pass consumption as $m$ is usually large in pre-training dataset. Please refer to Appendix 3 for the detailed derivation.

**Combined Adjustment**. We can combine feature-wise and class-wise adjustment to make the stratification in backdoor adjustment much more fine-grained. Our combination is simple: applying feature-wise adjustment after class-wise adjustment. Thus, we have:

$$ P(Y|do(X = \mathbf{x})) \approx \frac{1}{n} \sum_{i=1}^{n} P(Y|[\mathbf{x}]_c \oplus \frac{1}{m} \sum_{j=1}^{m} [P(a_j|\mathbf{x})\bar{\mathbf{x}}_j]_c), \text{ where } c = \{k|k \in \mathcal{F}_i \cap \mathcal{I}_t\}. \quad (4) $$

## 4 Related Work

**Few-Shot Learning**. FSL has a wide spectrum of methods, including fine-tuning [11, 16], optimizing model initialization [18, 40], generating model parameters [51, 34], learning a feature space for a better separation of sample categories [62, 72], feature transfer [54, 41], and transductive learning that additionally uses query set data [16, 27, 25]. Thanks to them, the classification accuracy has been drastically increased [27, 72, 68, 35]. However, accuracy as a single number cannot explain the paradoxical phenomenon in Figure 2. Our work offers an answer from a causal standpoint by showing that pre-training is a confounder. We not only further improve the accuracy of various FSL methods, but also explain the reason behind the improvements. In fact, the perspective offered by our work can benefit all the tasks that involve pre-training—any downstream task can be seen as FSL compared to the large-scale pre-training data.

**Negative Transfer**. The above phenomenon is also known as the negative transfer, where learning in source domain contributes negatively to the performance in target domain [42]. Many research works have being focused on when and how to conduct this transfer learning [28, 4, 76]. Yosinski *et al.* [69] split ImageNet according to man-made objects and natural objects as a test bed for feature transferability. They resemble the $\mathcal{S} \not\sim \mathcal{Q}$ settings used in Figure 2(a). Other work also revealed that using deeper backbone might lead to degraded performance when the domain gap between training and test is large [31]. Some similar findings are reported in the few-shot setting [47] and NLP tasks [58]. Unfortunately, they didn't provide a theoretical explanation why it happens.

**Causal Inference**. Our work aims to deal with the pre-training confounder in FSL based on causal inference [45]. Causal inference was recently introduced to machine learning [38, 7] and has been applied to various fields in computer vision. [67] proposes a retrospective for image captioning and other applications include image classification [10, 36], imitation learning [13], long-tailed recognition [56] and semantic segmentation [73]. We are the first to approach FSL from a causal perspective. We would like to highlight that data-augmentation based FSL can also be considered as approximated intervention. These methods learn to generate additional support samples with image deformation [12, 74] or generative models [2, 75]. This can be view as physical interventions on the image features. Regarding the causal relation between image $X$ and label $Y$, some works adopted anti-causal learning [39], *i.e.*, $Y \rightarrow X$, where the assumption is that labels $Y$ are disentangled enough to be treated as Independent Mechanism (IM) [43, 55], which generates observed images $X$ through $Y \rightarrow X$. However, our work targets at the more general case where labels can be entangled (*e.g.*"lion" and "dog" share the semantic "soft fur") and the IM assumption may not hold. Therefore, we use causal prediction $X \rightarrow Y$ as it is essentially a reasoning process, where the IM is captured by $D$, which is engineered to be disentangled through CNN (*e.g.*, the conv-operations are applied independently). In this way, $D$ generates visual features through $D \rightarrow X$ and emulates human's naming process through $D \rightarrow Y$ (*e.g.*, "fur", "four-legged"→ "meerkat"). In fact, the causal direction $X \rightarrow Y$ (NOT anti-causal $Y \rightarrow X$) has been empirically justified in complex CV tasks [30, 63, 56, 57].

## 5 Experiments

### 5.1 Datasets and Settings

**Datasets**. We conducted experiments on benchmark datasets in FSL literature: 1) ***mini*ImageNet** [62] containing 600 images per class over 100 classes. We followed the split proposed in [48]: 64/16/20

classes for train/val/test. 2) *tiered*ImageNet [49] is much larger compared to *mini*ImageNet with 608 classes and each class around 1,300 samples. These classes were grouped into 34 higher-level concepts and then partitioned into 20/6/8 disjoint sets for train/val/test to achieve larger domain difference between training and testing. 3) Caltech-UCSD Birds-200-2011 (**CUB**) [65] for cross-domain evaluation. It contains 200 classes and each class has around 60 samples. The models used for CUB test were trained on the *mini*ImageNet. Training and evaluation settings on *mini*ImageNet and *tiered*ImageNet are included in Appendix 5.

**Implementation Details**. We pre-trained the 10-layer ResNet (ResNet-10) [23] and the WideResNet (WRN-28-10) [70] as our backbones. Our proposed IFSL supports both fine-tuning and meta-learning. For fine-tuning, we applied average pooling on the last residual block and used the pooled features to train classifiers. For meta-learning, we deployed 5 representative methods that cover a large spectrum of meta-learning based FSL: 1) model initialization: MAML [18], 2) weight generator: LEO [51], transductive learning: SIB [27], 4) metric learning: MatchingNet (MN) [62], and 5) feature transfer: MTL [54]. For both fine-tuning and meta-learning, our IFSL aims to the learn classifier $P(Y|do(X))$ instead of the conventional $P(Y|X)$. Detailed implementations are given in Appendix 5.

**Evaluation Metrics**. Our evaluation is based on the following metrics: 1) Conventional accuracy (**Acc**) is the average classification accuracy commonly used in FSL [18, 62, 54]. 2) **Hardness-specific Acc**. For each query, we define a hardness that measures its semantic dissimilarity to the support set, and accuracy is then computed at different levels of query hardness. Specifically, query hardness is computed by $h = \log\big((1-s)/s\big)$ and $s = exp\langle\mathbf{r}^+, \mathbf{p}^+_{c=gt}\rangle/\sum_c exp\langle\mathbf{r}^+, \mathbf{p}^+_c\rangle$, where $\langle\cdot\rangle$ is the cosine similarity, $(\cdot)^+$ represents the ReLU activation function, $\mathbf{r}$ denotes the $\Omega$ prediction logits of query, $\mathbf{p}_c$ denotes the average prediction logits of class $c$ in the support set and $gt$ is the ground-truth of query. Using **Hardness-specific Acc** is similar to evaluating the hardness of FSL tasks [16], while ours is query-sample-specific and hence is more fine-grained. Later, we will show its effectiveness to unveil the spurious effects in FSL. 3) Feature localization accuracy (**CAM-Acc**) quantifies if a model "pays attention" to the actual object when making prediction. It is defined as the percentage of pixels inside the object bounding box by using Grad-CAM [53] score larger than 0.9. Compared to **Acc** that shows if the prediction is correct, **CAM-Acc** reveals whether the prediction is based on the correct visual cues.

Table 1: Acc (%) averaged over 2000 5-way FSL tasks before and after applying IFSL. We obtained the results by using official code and our backbones for a fair comparison across methods. We also implemented SIB in both transductive and inductive setting to facilitate fair comparison. For IFSL, we reported results of combined adjustment as it almost always outperformed feature-wise and class-wise adjustment. See Appendix 6 for Acc and 95% confidence intervals on all 3 types of adjustment.

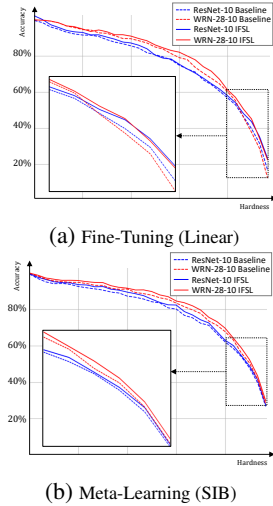

(a) Fine-Tuning (Linear)

(b) Meta-Learning (SIB)

Figure 5: Accuracy across query hardness on 5-shot fine-tuning and meta-learning. Additional results are shown in Appendix 6.

| | Method | | ResNet-10 | | | | WRN-28-10 | | | |
| | | | *mini*ImageNet | | *tiered*ImageNet | | *mini*ImageNet | | *tiered*ImageNet | |
| | | | 5-shot | 1-shot | 5-shot | 1-shot | 5-shot | 1-shot | 5-shot | 1-shot |
|---|---|---|---|---|---|---|---|---|---|---|
| **Fine-Tuning** | Linear | | 76.38 | 56.26 | 81.01 | 61.39 | 79.79 | 60.69 | 85.37 | 67.27 |
| | | +IFSL+2.19 | 77.97+1.59 | 60.13+3.87 | 82.08+1.07 | 64.29+2.9 | 80.97+1.18 | 64.12+3.43 | 86.19+0.82 | 69.96+2.69 |
| | Cosine | | 76.68 | 56.40 | 81.13 | 62.08 | 79.72 | 60.83 | 85.41 | 67.30 |
| | | +IFSL+1.77 | 77.63+0.95 | 59.84+3.44 | 81.75+0.62 | 64.47+2.39 | 80.74+1.02 | 63.76+2.93 | 86.13+0.72 | 69.36+2.06 |
| | *k*-NN | | 76.63 | 55.92 | 80.85 | 61.16 | 79.60 | 60.34 | 84.67 | 67.25 |
| | | +IFSL+3.13 | 78.42+1.79 | 62.31+6.36 | 81.98+1.13 | 65.71+4.55 | 81.08+1.48 | 64.98+4.64 | 86.06+1.39 | 70.94+3.69 |
| **Meta-Learning** | MAML [18] | | 70.85 | 56.59 | 74.02 | 59.17 | 73.92 | 58.02 | 77.20 | 61.40 |
| | | +IFSL+5.55 | 76.37+5.52 | 59.36+2.77 | 81.04+7.02 | 63.88+4.71 | 79.25+5.33 | 62.84+4.82 | 85.10+7.90 | 67.70+6.30 |
| | LEO [51] | | 74.49 | 58.48 | 80.25 | 65.25 | 75.86 | 59.77 | 82.15 | 68.90 |
| | | +IFSL+1.94 | 76.91+2.42 | 61.09+2.61 | 81.43+1.18 | 66.03+0.78 | 77.72+1.86 | 62.19+2.42 | 85.04+2.89 | 70.28+1.38 |
| | MTL [54] | | 75.65 | 58.49 | 81.14 | 64.29 | 77.30 | 62.99 | 83.23 | 70.08 |
| | | +IFSL+2.02 | 78.03+2.38 | 61.17+2.68 | 82.35+1.21 | 65.72+1.43 | 80.20+2.9 | 64.40+1.41 | 86.02+2.79 | 71.45+1.37 |
| | MN [62] | | 75.21 | 61.05 | 79.92 | 66.01 | 77.15 | 63.45 | 82.43 | 70.38 |
| | | +IFSL+1.34 | 76.73+1.52 | 62.64+1.59 | 80.79+0.87 | 67.30+1.29 | 78.55+1.40 | 64.89+1.44 | 84.03+1.60 | 71.41+1.03 |
| | SIB [27] (transductive) | | 78.88 | 67.10 | 85.09 | 77.64 | 81.73 | 71.31 | 88.19 | 81.97 |
| | | +IFSL+1.15 | 80.32+1.44 | 68.85+1.75 | 85.43+0.34 | 78.03+0.39 | 83.21+1.48 | 73.51+2.20 | 88.69+0.50 | 83.07+1.10 |
| | SIB [27] (inductive) | | 75.64 | 57.20 | 81.69 | 65.51 | 78.17 | 60.12 | 84.96 | 69.20 |
| | | +IFSL+2.05 | 77.68+2.04 | 60.33+3.13 | 82.75+1.06 | 67.34+1.83 | 80.05+1.88 | 63.14+3.02 | 86.14+1.18 | 71.45+2.25 |

## 5.2 Results and Analysis

**Conventional Acc**. 1) From Table 1, we observe that IFSL consistently improves fine-tuning and meta-learning in all settings, which suggests that IFSL is agnostic to methods, datasets, and backbones. 2) In particular, the improvements are typically larger on 1-shot than 5-shot. For example, in fine-

Table 2: Comparison with state-of-the-arts of 5-way 1-/5-shot Acc (%) on *mini*ImageNet and *tiered*ImageNet.

| Method | Backbone | *mini*ImageNet 5-shot | *mini*ImageNet 1-shot | *tiered*ImageNet 5-shot | *tiered*ImageNet 1-shot |
|---|---|---|---|---|---|
| Baseline++ [11] | ResNet-10 | 75.90 | 53.97 | - | - |
| IdeMe-Net[†] [12] | ResNet-10 | 73.78 | 57.61 | 80.34 | 60.32 |
| TRAML [32] | ResNet-12 | 79.54 | 67.10 | - | - |
| DeepEMD [72] | ResNet-12 | 82.41 | 65.91 | 86.03 | 71.16 |
| CTM [33] | ResNet-18 | 80.51 | 64.12 | 84.28 | 68.41 |
| FEAT [68] | WRN-28-10 | 81.80 | 66.69 | 84.38 | 70.41 |
| Tran. Baseline [16] | WRN-28-10 | 78.40 | 65.73 | 85.50 | 73.34 |
| wDAE-GNN [19] | WRN-28-10 | 78.85 | 62.96 | 83.09 | 68.18 |
| SIB[†] [27] | WRN-28-10 | 81.73 | 71.31 | 88.19 | 81.97 |
| **SIB+IFSL (ours)** | WRN-28-10 | **83.21** | **73.51** | **88.69** | **83.07** |

[†] Using our pre-trained backbone.

Table 3: Results of cross-domain evaluation: *mini*ImageNet → CUB. The whole report is in Appendix 6.

| Backbone | Method | | 5-shot | 1-shot |
|---|---|---|---|---|
| ResNet-10 | Linear | | 58.84 | 42.25 |
| | | +IFSL | 60.65 | 45.14 |
| | SIB | | 60.60 | 45.87 |
| | | +IFSL | 62.07 | 47.07 |
| WRN-28-10 | Linear | | 62.12 | 42.89 |
| | | +IFSL | 64.15 | 45.64 |
| | SIB | | 62.59 | 49.16 |
| | | +IFSL | 64.43 | 50.71 |

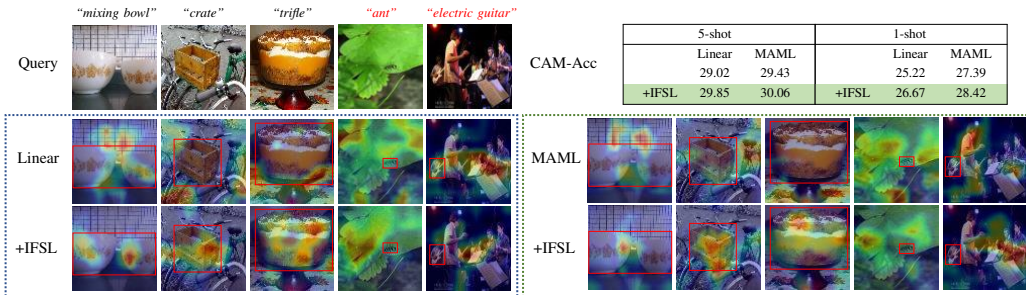

Figure 6: Some *mini*ImageNet visualizations of Grad-Cam [53] activation of query images and the CAM-Acc (%) table of using linear classifier and MAML. Categories with red text represent failed cases. The complete results on CAM-Acc are shown in Appendix 6, where IFSL achieves similar or better results in all settings.

tuning, the average performance gain is 1.15% on 5-shot and 3.58% on 1-shot. The results support our analysis in Section 2.3 that FSL models are more prone to bias in lower-shot settings. 3) Regarding the average improvements on fine-tuning vs. meta-learning (*e.g.* $k$-NN and MN), we observe that IFSL improves more on fine-tuning in most cases. We conjecture that this is because meta-learning is an implicit form of intervention, where randomly sampled meta-training episodes effectively stratify the pre-trained knowledge. This suggests that meta-learning is fundamentally superior over fine-tuning due to increased robustness against confounders. We will investigate this potential theory in future work. 4) Additionally we see that the improvements on *mini*ImageNet are usually larger than that on *tiered*ImageNet. A possible reason is the much larger training set for *tiered*ImageNet: it substantially increases the breadth of the pre-trained knowledge and the resulting models explain query samples much better. 5) According to Table 1 and Table 2, it is clear that our $k$-NN+IFSL outperforms IdeMe-Net [12] using the same pre-trained ResNet-10. This shows that using data augmentation — a method of physical data intervention as in IdeMe-Net [12] is inferior to our causal intervention in IFSL. 6) Overall, our IFSL achieves the new state-of-the-art on both datasets. Note that IFSL is flexible to be plugged into different baselines.

**Hardness-specific Acc.** 1) Figure 5(a) shows the plot of Hardness-specific Acc of fine-tuning. We notice that when query becomes harder, ResNet-10 (blue curves) becomes superior to WRN-28-10 (red curves). This tendency is consistent with Figure 2(a) illustrating the effect of the confounding bias caused by pre-training. 2) Intriguingly, in Figure 5(b), we notice that this tendency is reversed for meta-learning, *i.e.*, deeper backbone always performs better. The improved performance of deeper backbone on hard queries suggests that meta-learning should have some functions to remove the confounding bias. This evidence will inspire us to provide a causal view of meta-learning in future work. 3) Overall, Figure 5 shows that using IFSL futher improves fine-tuning and meta-learning consistently across all hardness, validating the effectiveness of the proposed causal intervention.

**CAM-Acc & Visualization.** In Figure 6, we compare +IFSL to baseline linear classifier on the left and to baseline MAML [18] on the right, and summarize CAM-Acc results in the upper-right table. From the visualization, we see that using IFSL let the model pay more attention to the objects. However, notice that all models failed in the categories colored as red. A possible reason behind the failures is the extremely small size of the object — models have to resort to context for prediction. From the numbers, we can see our improvements for 1-shot are larger than that for 5-shot, consistent

with our findings using other evaluation metrics. These results suggest that IFSL helps models use the correct visual semantics for prediction by removing the confounding bias.

**Cross-Domain Generalization Ability.** In Table 3, we show the testing results on CUB using the models trained on the *mini*ImageNet. The setting is challenging due to the big domain gap between the two datasets. We chose linear classifier as it outperforms cosine and $k$-NN in cross-domain setting and compared with transductive method — SIB. The results clearly show that IFSL works well in this setting and brings consistent improvements, with the average 1.94% of Acc. In addition, we can see that applying IFSL brings larger improvements to the inductive linear classifier than to the transductive SIB. It is possibly because transductive methods involve unlabeled query data and performs better than inductive methods with the additional information. Nonetheless we observe that IFSL can further improve SIB in cross-domain (Table 3) and single-domain (Table 1) generalization.

## 6 Conclusions

We presented a novel casual framework: Interventional Few-Shot Learning (IFSL), to address an overlooked deficiency in recent FSL methods: the pre-training is a confounder hurting the performance. Specifically, we proposed a structural causal model of the causalities in the process of FSL and then developed three practical implementations based on the backdoor adjustment. To better illustrate the deficiency, we diagnosed the classification accuracy comprehensively across query hardness, and showed that IFSL improves all the baselines across all the hardness. It is worth highlighting that the contribution of IFSL is not only about improving the performance of FSL, but also offering a causal explanation why IFSL works well: it is a causal approximation to many-shot learning. We believe that IFSL may shed light on exploring the new boundary of FSL, even though FSL is well-known to be ill-posed due to insufficient data. To upgrade IFSL, we will seek other observational intervention algorithms for better performance, and devise counterfactual reasoning for more general few-shot settings such as domain transfer.

## 7 Acknowledgements

The authors would like to thank all the anonymous reviewers for their constructive comments and suggestions. This research is partly supported by the Alibaba-NTU Singapore Joint Research Institute, Nanyang Technological University (NTU), Singapore; the Singapore Ministry of Education (MOE) Academic Research Fund (AcRF) Tier 1 and Tier 2 grant; and Alibaba Innovative Research (AIR) programme. We also want to thank Alibaba City Brain Group for the donations of GPUs.

## 8 Broader Impact

The proposed method aims to improve the Few-Shot Learning task. Advancements in FSL helps the deployment of machine learning models in areas where labelled data is difficult or expensive to obtain and it is closely related to social well-beings: few-shot drug discovery or medical imaging analysis in medical applications, cold-start item recommendation in e-commerce, few-shot reinforcement learning for industrial robots, *etc.*. Our method is based on causal inference and the analysis is rooted on causation rather than correlation. The marriage between causality and machine learning can produce more robust, transparent and explainable models, broadening the applicability of ML models and promoting fairness in artificial intelligence.

## Footnotes

[1]In causal graph, the junction $A \to B \leftarrow C$ is called a "collider", making $A$ and $C$ independent even though $A$ and $C$ are linked via $B$ [44]. For example, $A$ = "Quality", $C$ = "Luck", and $B$ = "Paper Acceptance".

[2]We assume that the combinations of the feature dimensions or classes are linear, otherwise the adjustment requires prohibitive $\mathcal{O}(2^n)$ sampling. We will relax this assumption in future work.

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
