[Supplementary Material]

# Supplementary Material for Interventional Few-Shot Learning

**Zhongqi Yue**[1,3], **Hanwang Zhang**[1], **Qianru Sun**[2], **Xian-Sheng Hua**[3]

[1]Nanyang Technological University, [2]Singapore Management University, [3]Alibaba Group
yuez0003@ntu.edu.sg, hanwangzhang@ntu.edu.sg,
qianrusun@smu.edu.sg, xiansheng.hxs@alibaba-inc.com

This supplementary material is organized as follows:

- Section A.1 details our analysis in Section 2.3 by showing many-shot learning converges to true causal effect through instrumental variable (IV);

- Section A.2 gives the derivation for the backdoor adjustment formula in Eq. (1);

- Section A.3 presents the detailed derivation for the NWGM approximation used in Eq. (3) and (4);

- Section A.4 includes the algorithms for adding IFSL to fine-tuning and meta-learning;

- Section A.5 shows the implementation details for pre-training (Section A.5.1), fine-tuning (Section A.5.2) and meta-learning (Section A.5.3);

- Section A.6 includes additional experimental results on Conventional Acc (Section A.6.1), Hardness-Specific Acc (Section A.6.2), CAM-Acc (Section A.6.3) and cross-domain evaluation (Section A.6.4).

## A.1   Instrumental Variable

In this section, we will show that in our causal graph for many-shot learning, the sampling ID $I$ is essentially an instrumental variable for $X \to Y$ that achieves $P(Y|I) \approx P(Y|do(X))$. Before introducing instrumental variable, we first formally define *d-separation* [7], which gives a criterion to study the dependencies between nodes (data variables) in any structural causal model.

**d-separation**. A set of nodes $Z$ blocks a path $p$ if and only if 1) $p$ contains a *chain* $A \to B \to C$ or a *fork* $A \leftarrow B \to C$ and the middle node $B$ is in $Z$; 2) $p$ contains a *collider* $A \to B \leftarrow C$ such that the middle node $B$ and its descendants are not in $Z$. If conditioning on $Z$ blocks every path between $X$ and $Y$, we say $X$ and $Y$ are *d-separated* conditional on $Z$, *i.e.*, $X$ and $Y$ are independent given $Z$ ($X \perp\!\!\!\perp Y|Z$).

**Instrumental Variable**. For a structual causal model $\mathcal{G}$, a variable Z is an *instrumental variable* (IV) to $X \to Y$ by satisfying the graphical criteria [9]: 1) $(Z \perp\!\!\!\perp Y)_{\mathcal{G}_{\overline{X}}}$ ; 2) $(Z \not\!\perp\!\!\!\perp X)_{\mathcal{G}}$ , where $\mathcal{G}_{\overline{X}}$ is the manipulated graph where all incoming arrows to node $X$ are deleted. For the SCM of many-shot learning in Figure 4(a), it is easy to see that $I$ satisfies both criteria and therefore it is an IV for $X \to Y$. However, in the few-shot SCM in Figure 4(b), the paths $I \leftarrow X \leftarrow D \to C \to Y$ and $I \leftarrow X \to C \to Y$ are not blocked in $\mathcal{G}_{\overline{X}}$, which means the first criterion is not met $(I \not\!\perp\!\!\!\perp Y)_{\mathcal{G}_{\overline{X}}}$ and $I$ is not an instrumental variable in the few-shot learning case.

Instrumental variable can help find the true causal effect even in the presence of confounder. This is due to the collider junction that makes the IV and confounder independent ($I \perp\!\!\!\perp D$ in Figure 4(a)). To see this, we will first consider a simplified case of Figure 4(a) where each causal link represents a linear relationship and we aim to find the true causal effect from $X \to Y$ through linear regression. Without loss of generality, let $I, X, Y$ take the value of real number. Denote $r_{IX}, r_{XY}$, and $r_{IY}$ as the slope of regression line between $I$ and $X$, $X$ and $Y$, $I$ and $Y$ respectively. Notice

that $r_{XY}$ is spurious as it is contaminated by the backdoor path $X \leftarrow D \rightarrow C \rightarrow Y$. However, since the path $I \rightarrow X \leftarrow D \rightarrow C \rightarrow Y$ is blocked due to collider at $X$, $r_{IY}$ is free from confounding bias. Therefore $r_{IY}/r_{IX}$ gives the true causal effect from $X \rightarrow Y$. Similarly, in the classification case of many-shot learning, a classifier is trained to maximize the conditional probability on the IV $P(Y|I)$. As the ID-sample matching $I \rightarrow X$ is deterministic, the classifier eventually learns to predict based on the true causal relationship $X \rightarrow Y$. Yet in the complex case of image classification, it is unreasonable to assume linear relationships between variables. In the nonlinear case, it is shown in [2] that observations on IV provide a bound for the true causal effect. This means that learning based on $P(Y|I)$ provides an approximation to the true causal effect, $i.e. P(Y|I) \approx P(Y|do(X))$.

## A.2 Derivation of Backdoor Adjustment for the Proposed Causal Graph

We will show the derivation of the backdoor adjustment for the causal graph in Figure 3(a) using the three rules of *do*-calculus [8].

For a causal directed acyclic graph $\mathcal{G}$, let $X, Y, Z$ and $W$ be arbitrary disjoint sets of nodes. We use $\mathcal{G}_{\overline{X}}$ to denote the manipulated graph where all incoming arrows to node $X$ are deleted. Similarly $\mathcal{G}_{\underline{X}}$ represents the graph where outgoing arrows from node $X$ are deleted. We use lower case $x, y, z$ and $w$ for specific values taken by each set of nodes: $X = x, Y = y, Z = z$ and $W = w$. For any interventional distribution compatible with $\mathcal{G}$, we have the following three rules:

**Rule 1** Insertion/deletion of observations:

$$P(y|do(x), z, w) = P(y|do(x), w), \text{if} (Y \perp\!\!\!\perp Z | X, W)_{\mathcal{G}_{\overline{X}}} \tag{A1}$$

**Rule 2** Action/observation exchange:

$$P(y|do(x), do(z), w) = P(y|do(x), z, w), \text{if} (Y \perp\!\!\!\perp Z | X, W)_{\mathcal{G}_{\overline{X}\underline{Z}}} \tag{A2}$$

**Rule 3** Insertion/deletion of actions:

$$P(y|do(x), do(z), w) = P(y|do(x), w), \text{if} (Y \perp\!\!\!\perp Z | X, W)_{\mathcal{G}_{\overline{X}\overline{Z(W)}}}, \tag{A3}$$

where $Z(W)$ is the set of nodes in $Z$ that are not ancestors of any $W$-node in $\mathcal{G}_{\overline{X}}$.

In our causal graph, the desired interventional distribution $P(Y|do(X = \mathbf{x}))$ can be derived by:

$$P(Y|do(\mathbf{x})) = \sum_d P(Y|do(X = \mathbf{x}), D = d)P(D = d|do(X = \mathbf{x})) \tag{A4}$$

$$= \sum_d P(Y|do(X = \mathbf{x}), D = d)P(D = d) \tag{A5}$$

$$= \sum_d P(Y|X = \mathbf{x}, D = d)P(D = d) \tag{A6}$$

$$= \sum_d \sum_c P(Y|X = \mathbf{x}, D = d, C = c)P(C = c|X = \mathbf{x}, D = d)P(D = d) \tag{A7}$$

$$= \sum_d P(Y|X = \mathbf{x}, D = d, C = g(\mathbf{x}, d))P(D = d), \tag{A8}$$

where Eq. (A4) and Eq. (A7) follow the law of total probability; Eq. (A5) uses Rule 3 given $D \perp\!\!\!\perp X$ in $\mathcal{G}_{\overline{X}}$; Eq. (A6) uses Rule 2 to change the intervention term to observation as $(Y \perp\!\!\!\perp X | D)$ in $\mathcal{G}_{\underline{X}}$; Eq. (A8) is because in our causal graph, $C$ takes a deterministic value given by function $g(\mathbf{x}, d)$. This reduces summation over all values of $C$ in Eq. (A7) to a single probability measure in Eq. (A8).

## A.3 Derivation of NWGM Approximation

We will show the derivation of NWGM approximation used in Eq. (3) and (4). In a $K$-way FSL problem, let $f(\cdot)$ be a classifier function that calculates logits for $K$ classes and $\sigma$ be the softmax function over $K$ classes. The approximation effectively moves the outer expectation inside the classifier function: $\mathbb{E}[\sigma(f(\cdot))] \approx \sigma(f(\mathbb{E}[\cdot]))$.

We will first show the derivation for moving the expectation inside softmax function, *i.e.*, $\mathbb{E}[\sigma(f(\cdot))] \approx \sigma(\mathbb{E}[f(\cdot)])$. Without loss of generality, the backdoor adjustment formula in Eq. (3) and Eq. (4) can be written as:

$$P(Y = y | do(X = \mathbf{x})) = \sum_{d \in D} \sigma(f_y(\mathbf{x} \oplus \mathbf{c})) P(d), \tag{A9}$$

where $D$ represents the set of stratifications, $f_y$ is the classifier logit for class $y$, $\mathbf{c} = g(\mathbf{x}, d)$ is the feature concatenated to $\mathbf{x}$ in Eq. (3) and (4) and $P(d)$ is the prior for each stratificaction.

It is shown in [1] that Eq. (A9) can be approximated by the Normalized Weighted Geometric Mean (NWGM) as:

$$\sum_{d \in D} \sigma(f_y(\mathbf{x} \oplus \mathbf{c})) P(d) \approx NWGM_{d \in D}(\sigma(f_y(\mathbf{x} \oplus \mathbf{c}))) \tag{A10}$$

$$= \frac{\prod_d [exp(f_y(\mathbf{x} \oplus \mathbf{c}))]^{P(d)}}{\sum_{i=1}^K \prod_d [exp(f_i(\mathbf{x} \oplus \mathbf{c}))]^{P(d)}} \tag{A11}$$

$$= \frac{exp(\sum_d f_y(\mathbf{x} \oplus \mathbf{c}) P(d))}{\sum_{i=1}^K exp(\sum_d f_i(\mathbf{x} \oplus \mathbf{c}) P(d))} \tag{A12}$$

$$= \sigma\left(\mathbb{E}_d[f_y(\mathbf{x} \oplus \mathbf{c})]\right), \tag{A13}$$

where Eq. (A10) follows [1], Eq. (A11) follows the definition of NWGM, Eq. (A12) is because $exp(a)^b = exp(ab)$.

Next we will show the derivation for linear, cosine and $k$-NN classifier to further move the expectation inside the classifier function, *i.e.*, $\sigma(\mathbb{E}[f(\cdot)]) = \sigma(f(\mathbb{E}[\cdot]))$.

For the linear classifier, $f(\mathbf{x} \oplus \mathbf{c}) = \mathbf{W}_1\mathbf{x} + \mathbf{W}_2\mathbf{c}$, where $\mathbf{W}_1, \mathbf{W}_2 \in \mathbb{R}^{K \times N}$ denote the learnable weight, $N$ is the feature dimension, which is the same for $\mathbf{x}$ and $\mathbf{c}$ in Eq. (3) and (4). The bias term is dropped as it does not impact our analysis. Now the expectation can be further moved inside the classifier function through:

$$\sum_d f(\mathbf{x} \oplus \mathbf{c})) P(d) = \sum_d (\mathbf{W}_1\mathbf{x} + \mathbf{W}_2\mathbf{c}) P(d) \tag{A14}$$

$$= \mathbf{W}_1\mathbf{x} + \sum_d \mathbf{W}_2\mathbf{c} P(d) \tag{A15}$$

$$= f(\mathbf{x} \oplus \sum_d \mathbf{c} P(d)), \tag{A16}$$

where Eq. (A15) is because the feature vector $\mathbf{x}$ is the same for all $d$ and $\mathbb{E}_d[\mathbf{x}] = \mathbf{x}$.

For the cosine classifier, $f(\mathbf{x} \oplus \mathbf{c}) = (\mathbf{W}_1\mathbf{x} + \mathbf{W}_2\mathbf{c}) / \|\mathbf{x} \oplus \mathbf{c}\| \|\mathbf{W}\|$, where $\mathbf{W} \in \mathbb{R}^{K \times 2N}$ is the concatenation of $\mathbf{W}_1$ and $\mathbf{W}_2$. In the special case where $\mathbf{x}$ and $\mathbf{c}$ are unit vector, $\|\mathbf{x} \oplus \mathbf{c}\|$ is $\sqrt{2}$ and the cosine classifier function reduces to a linear combination of terms involving only $\mathbf{x}$ and only $\mathbf{c}$. From there, the analysis for linear classifier follows and we have $\sigma(\mathbb{E} f(\cdot)) = \sigma(f(\mathbb{E} \cdot))$ for cosine classifier. In the general case where $\mathbf{x}$ and $\mathbf{c}$ are not unit vector, moving the expectation inside cosine classifier function is an approximation $\sigma(\mathbb{E}[f(\cdot)]) \approx \sigma(f(\mathbb{E}[\cdot]))$.

For the $k$-NN classifier, our implementation calculates class centroids using the mean feature of the $K$ support sets and then uses the nearest centroid for prediction (1-NN). Specifically, let $\mathbf{x}$ be a feature vector and $\mathbf{x}'$ be the $i$th class centroid, $i \in \{1, \ldots, K\}$. The logit for class $i$ is given by $f_i(\mathbf{x}) = - \|\mathbf{x} - \mathbf{x}'\|^2$. It is shown in [11] that $k$-NN classifier that uses squared Euclidean distance to generate logits is equivalent to a linear classifier with a particular parameterization. Therefore, our analysis on linear classifier follows for $k$-NN.

In summary, the derivation of $\mathbb{E}[\sigma(f(\cdot))] \approx \sigma(f(\mathbb{E}[\cdot]))$ is a two-stage process where we first move the expectation inside the softmax function and then further move it inside the classifier function.

## A.4 Algorithms for Fine-tuning and Meta-Learning with IFSL

In this section, we will briefly revisit the settings of fine-tuning and meta-learning and introduce how to integrate IFSL into them.

In fine-tuning, the goal is to train a classifier $\theta$ conditioned on the current support set $\mathcal{S} = \{(\mathbf{x}_i, y_i)\}_{i=1}^{n_s}$, where $\mathbf{x}_i$ is the feature generated by $\Omega$ for $i$th sample, $y_i$ is the ground-truth label for $i$th sample and $n_s$ is the support set size. This is achieved by first predicting the support label $\hat{y}$ using the classifier $P(y|\mathbf{x};\theta)$. Then with the predicted label $\hat{y}$ and ground-truth label $y$, one can calculate a loss $\mathcal{L}(\hat{y}, y)$ (usually cross-entropy loss) to update the classifier parameter, *e.g.* through stochastic gradient descent. Adding IFSL to fine-tuning is simple: 1) Pick an adjustment strategy introduced in Section 3. Each implementation defines the set of pre-trained knowledge stratifications $D$, function form of $g(X, D)$, function form of $P(Y|X, D, C)$ and the prior $P(D)$; 2) The classifier prediction is now based on $P(Y|do(X);\theta)$. The process of fine-tuning with IFSL is summarized in Algorithm 1. Note that for the non-parametric $k$-NN classifier, the fine-tuning process is not applicable. When adding IFSL to $k$-NN, each sample is represented by the *adjusted feature* instead of original feature $\mathbf{x}$. Please refer to the classifier inputs in Eq. (2), (3) and (4) for the exact form of adjusted feature.

In meta-learning, the goal is to learn the additional "learning behavior" parameterized by $\phi$ using training episodes $\{(\mathcal{S}_i, \mathcal{Q}_i)\}$ sampled from training dataset $\mathcal{D}$. The classifier in meta-learning makes predictions by additionally conditioning on the learning behavior, written as $P_\phi(y|\mathbf{x};\theta)$. Within each episode, $\theta$ is first fine-tuned on the support set $\mathcal{S}_i$. Then the fine-tuned model is tested on the query set $\mathcal{Q}_i$ to obtain the loss $\mathcal{L}_\phi(\mathcal{S}_i, \mathcal{Q}_i)$ (*e.g.* using cross-entropy loss). Finally the loss is used to update $\phi$ using an optimizer. It is also easy to integrate IFSL into meta-learning by only changing the classifier from $P_\phi(y|\mathbf{x};\theta)$ to $P_\phi(y|do(\mathbf{x});\theta)$. The flow of meta-learning with IFSL is presented in Algorithm 2. Firstly notice that the initialization of $\theta$ in each task may depend on $\phi$ or $\mathcal{S}_i$. For example, in MAML [4] $\phi$ essentially defines an initialization of model parameters, and in LEO [10] the initial classifier parameter is generated conditioned on $\phi$ and $\mathcal{S}_i$. Secondly, although the fine-tuning of $\theta$ largely follows Algorithm 1, some meta-learning methods additionally utilize meta-knowledge $\phi$. For example, in SIB the gradients for updating $\theta$ are predicted by $\phi$ using unlabelled query features instead of calculated from $\mathcal{L}(\hat{y}, y)$ as in Algorithm 1.

---

**Algorithm 1:** Fine-tuning + IFSL

**Input** : $D$, Support set $\mathcal{S} = \{(\mathbf{x}_i, y_i)\}_{i=1}^{n_s}$
**Output:** Fine-tuned classifier parameters $\theta$
Initialize $\theta$;
**while** *not converged* **do**
   **for** $i = 1, \ldots, n_s$ **do**
      **for** $d \in D$ **do**
         Calculate $c = g(\mathbf{x}, d)$;
         Obtain $P(Y|\mathbf{x}_i, c, d; \theta), P(d)$
      Prediction $\hat{y}_i = P(y|do(\mathbf{x}); \theta)$;
      Update $\theta$ using $\mathcal{L}(\hat{y}_i, y_i)$
**return** $\theta$

---

**Algorithm 2:** Meta-Learning + IFSL

**Input** : $D$, training dataset $\mathcal{D}$
**Output:** Optimized meta-parameters $\phi$
Initialize $\phi$;
**while** *not converged* **do**
   Sample $(\mathcal{S}_i, \mathcal{Q}_i)$ from $\mathcal{D}$ ;
   Initialize classifier $\theta$ with $\phi, \mathcal{S}_i$;
   Fine-tune $\theta$ using **Algorithm 1**
    conditioned on $\phi$ ;
   Predict query based on $P_\phi(y|do(\mathbf{x}); \theta)$;
   Update $\phi$ using $\mathcal{L}_\phi(\mathcal{S}_i, \mathcal{Q}_i; \theta)$
**return** $\phi$

---

## A.5 Implementation Details

### A.5.1 Pre-training

Prior to fine-tuning or meta-learning, we pre-trained a deep neural network (DNN) as feature extractor on the train split of a dataset. We use ResNet-10[5] or WRN-28-10[15] as feature extractor backbone. This section will present the architecture and exact training procedure for our backbones.

**Network Architecture.** The architecture of our ResNet-10 and WRN-28-10 backbone is shown in Figure A1. Specifically, each convolutional layer is described as "$n \times n$ conv, $p$", where $n$ is the kernel size and $p$ is the number of output channels. Convolutional layers with "/2" have a stride of 2 and are used to perform downsampling. The solid curved lines represent identity shortcuts, and the dotted lines are projection shortcuts implemented by $1 \times 1$ convolutions. The batch normalization and ReLU layers are omitted in Figure A1 to highlight the key structure of the two backbones.

**Pre-training Procedure.** The networks are trained from scratch with stochastic gradient descent in a fully-supervised manner, *i.e.*, minimizing cross-entropy loss on the train split of a dataset. Specifically the training is conducted on 90 epochs with early stopping using validation accuracy. We used batch

size of 256 and image size of $84 \times 84$. For data augmentation, a random patch is sampled from an image, resized to $84 \times 84$ and randomly flipped along horizontal axis before used for training. The initial learning rate is set to 0.1 and it is scaled down by factor of 10 every 30 epochs. We used the feature pre-processing methods in [14].

### A.5.2   Fine-Tuning

We consider linear, cosine and $k$-NN classifier for our fine-tuning experiments. In a $K$-way FSL problem, the detailed implementations for the classifier function $f(\mathbf{x})$ are:

**Linear**. $f(\mathbf{x}) = \mathbf{W}\mathbf{x} + \mathbf{b}$, where $\mathbf{x}$ is the input feature, $\mathbf{W} \in \mathbb{R}^{K \times N}$ is the learnable weight parameter, $N$ is the feature dimension and $\mathbf{b} \in \mathbb{R}^K$ is the learnable bias parameter.

**Cosine**. $f(\mathbf{x}) = \mathbf{W}\mathbf{x}/ \|\mathbf{W}\| \|\mathbf{x}\|$, where $\mathbf{W} \in \mathbb{R}^{K \times N}$ is the learnable weight parameter. We implemented cosine classifier without using the bias term.

$k$**-NN**. Our implementation of $k$-NN is similar to [11, 14]. For each of the $K$ classes, we first calculated the average support set feature (centroid) denoted as $\mathbf{x}_i, i \in \{1, \ldots, K\}$. The classifier output for class $i$ is then given by $f_i(\mathbf{x}) = -\|\mathbf{x} - \mathbf{x}_i\|^2$. Notice that the prediction given by this classifier will be the nearest centroid.

Figure A1: The architecture of our backbones: (a) ResNet-10 [5]; (b) WRN-28-10 [15].

We froze the backbone and used the average pooling layer output of $\Omega$ to learn the classifier. The output logits from classifier functions are normalized using softmax to generate probability output $P(y|\mathbf{x})$. For linear and cosine classifier, we followed [3] and trained the classifier for 100 iteration with a batch size of 4. For fine-tuning baseline, we set the learning rate as $1 \times 10^{-2}$ and weight decay as $1 \times 10^{-3}$. For IFSL, we set the learning rate as $5 \times 10^{-3}$ and weight decay as $1 \times 10^{-3}$. $k$-NN classifier is non-parametric and can be initialized directly from support set.

### A.5.3   Meta-Learning

**MAML.** MAML [4] aims to learn an initialization of network parameters such that it can be fine-tuned within a few steps to solve a variety of few-shot classification tasks. When using pre-trained network with MAML, it has been shown that learning initialization of the backbone can lead to unsatisfactory performance [3, 12]. Therefore in our experiment, we froze the backbone and appended a 2-layer MLP with ReLU activation in between the hidden layers and a linear classifier after the average pooling layer of $\Omega$. The hidden dimension of the layers in MLP is the same as output dimension of $\Omega$ (512 for ResNet-10 and 640 for WRN-28-10). The initialization of MLP and the linear classifier is meta-learnt using MAML. For hyper-parameters, we set the inner loop learning rate $\alpha = 0.01$, the outer loop learning rate $\beta = 0.01$ and the number of adaptation steps as 20. For IFSL, we adopted the same hyper-parameter setting and set $n=8$ for feature-wise and combined adjustment. Implementation-wise, we adopted the released code[1] from [3] and performed experiments on MAML without using first-order approximation. Following the implementation in [3], the model was trained on 10,000 randomly sampled tasks with model selection using validation accuracy. We used 2,000 randomly sampled tasks for validation and testing.

**MTL.** MTL [12] learns scaling and shifting parameters at each convolutional layer of the backbone. We used the MTL implementation released by the author[2] which adopts linear classifier. We integrated our ResNet-10 and WRN-28-10 backbones into the released code. The learning rate for scaling and shifting weights $\phi_{SS}$ and initial classifier parameters was set to $1 \times 10^{-4}$ uniformly. We set the inner loop learning rate for classifier as $1 \times 10^{-2}$ and the inner loop update step as 100. For IFSL, we

adopted the same hyper-parameter setting and set $n$=8 for feature-wise and combined adjustment. We trained the MTL model on 10,000 randomly sampled tasks with model selection using validation accuracy and used 2,000 randomly sampled tasks for validation and testing. We used 3 RTX 2080 Ti for MTL experiments on WRN-28-10 backbone.

**LEO.** LEO [10] learns to generate classifier parameters conditioned on support set and the generated parameters are further fine-tuned within each FSL task. Our experiments were conducted on the released code of LEO[3] using linear classifier. Following author's implementation, we saved the center cropped features from our pre-trained backbones and used the saved features to train LEO. For baseline, we used the hyper-parameter settings released by the author. For IFSL, we set $n$=8 for feature-wise and combined adjustment and halved the outer loop learning rate compared to baseline. The model was trained up to 100,000 randomly sampled tasks from training split with early stopping using validation accuracy. We used 2,000 randomly sampled tasks for validation and testing.

**Matching Net.** Matching Net (MN) [13] is a metric-based method that learns a distance kernel function for $k$-NN. We used the Matching Net implementation in [3]. The implementation follows the setup in [13] and uses LSTM-based fully conditional embedding. We set the learning rate as 0.01 uniformly. For IFSL, we used $n$=16 for feature-wise and combined adjustment. The model was trained using 10,000 randomly sampled tasks with model selection using validation accuracy. We used 2,000 randomly sampled tasks for validation and testing.

**SIB.** SIB [6] initializes classifier from support set and generates gradients conditioned on unlabelled query set features to update classifier parameters. We followed the SIB implementation released by the author[4] which uses cosine classifier. In the transductive setting, the query set size is set to 15. In the inductive setting, we used only 1 query sample randomly selected from the $K$ classes in each episode. In terms of hyper-parameter settings, we took 3 synthetic gradient steps ($K = 3$) for all our experiments. For baseline, the learning rate for SIB network and classifier was set to $1 \times 10^{-3}$ following author's implementation. For IFSL, we set the learning rate to $5 \times 10^{-4}$ and used $n$=4 for feature-wise and combined adjustment. In both transductive and inductive settings, we meta-trained SIB using 50,000 randomly sampled tasks with model selection using validation accuracy. We used 2,000 randomly sampled tasks for validation and testing.

## A.6   Additional Results

In this section, we include additional results on 1) **Conventional Acc** in Table A1 supplementary to Table 1; 2) **Hardness-Specific Acc** in Figure A2 for *mini*ImageNet and Figure A3 for *tiered*ImageNet, supplementary to Figure 5; 3) **CAM-Acc** in Table A2 supplementary to Figure 6; 4) **Cross-Domain Evaluation** in Table A3 supplementary to Table 3.

### A.6.1 Conventional Acc

Table A1: Supplementary to Table 1. Acc (%) and 95% confidence intervals averaged over 2000 5-way FSL tasks before and after applying three proposed implementations of adjustment. Specifically, "*ft*" refers to feature-wise adjustment, "*cl*" refers to class-wise adjustment and "*ft+cl*" refers to combined adjustment.

| | Method | | ResNet-10 | | | | WRN-28-10 | | | |
| | | | *mini*ImageNet | | *tiered*ImageNet | | *mini*ImageNet | | *tiered*ImageNet | |
| | | | 5-shot | 1-shot | 5-shot | 1-shot | 5-shot | 1-shot | 5-shot | 1-shot |
|---|---|---|---|---|---|---|---|---|---|---|
| **Fine-Tuning** | Linear | | 76.38 ± 0.36 | 56.26 ± 0.47 | 81.01 ± 0.38 | 61.39 ± 0.47 | 79.79 ± 0.33 | 60.69 ± 0.45 | 85.37 ± 0.34 | 67.27 ± 0.49 |
| | | *ft* | 76.84 ± 0.36 | 57.37 ± 0.43 | 81.45 ± 0.38 | 61.88 ± 0.47 | 80.22 ± 0.31 | 60.84 ± 0.45 | 85.70 ± 0.33 | 67.94 ± 0.48 |
| | | *cl* | 77.23 ± 0.34 | 59.45 ± 0.45 | 81.33 ± 0.38 | 62.60 ± 0.48 | 80.27 ± 0.32 | 62.15 ± 0.44 | 85.54 ± 0.33 | 68.11 ± 0.48 |
| | | *ft+cl* | 77.97 ± 0.34 | 60.13 ± 0.45 | 82.08 ± 0.37 | 64.29 ± 0.48 | 80.97 ± 0.31 | 64.12 ± 0.44 | 86.19 ± 0.34 | 69.96 ± 0.46 |
| | Cosine | | 76.68 ± 0.36 | 56.40 ± 0.46 | 81.13 ± 0.39 | 62.08 ± 0.47 | 79.72 ± 0.33 | 60.83 ± 0.46 | 85.41 ± 0.34 | 67.30 ± 0.50 |
| | | *ft* | 76.83 ± 0.35 | 56.86 ± 0.44 | 81.34 ± 0.37 | 62.45 ± 0.47 | 79.80 ± 0.32 | 61.25 ± 0.44 | 85.74 ± 0.33 | 67.86 ± 0.46 |
| | | *cl* | 76.99 ± 0.35 | 57.65 ± 0.45 | 81.42 ± 0.38 | 63.37 ± 0.48 | 79.96 ± 0.32 | 62.04 ± 0.45 | 85.77 ± 0.33 | 68.45 ± 0.46 |
| | | *ft+cl* | 77.63 ± 0.34 | 59.84 ± 0.46 | 81.75 ± 0.38 | 64.47 ± 0.48 | 80.74 ± 0.32 | 63.76 ± 0.45 | 86.13 ± 0.33 | 69.36 ± 0.47 |
| | *k*-NN | | 76.63 ± 0.36 | 55.92 ± 0.46 | 80.85 ± 0.39 | 61.16 ± 0.48 | 79.60 ± 0.32 | 60.34 ± 0.45 | 84.67 ± 0.34 | 67.25 ± 0.52 |
| | | *ft* | 77.98 ± 0.34 | 60.71 ± 0.44 | 81.95 ± 0.36 | 65.66 ± 0.48 | 81.17 ± 0.31 | 64.87 ± 0.44 | 85.76 ± 0.34 | 71.00 ± 0.47 |
| | | *cl* | 78.36 ± 0.35 | 61.32 ± 0.45 | 81.93 ± 0.37 | 65.71 ± 0.48 | 80.61 ± 0.31 | 64.43 ± 0.45 | 85.90 ± 0.33 | 70.08 ± 0.48 |
| | | *ft+cl* | 78.42 ± 0.34 | 62.31 ± 0.44 | 81.98 ± 0.38 | 65.71 ± 0.47 | 81.08 ± 0.32 | 64.98 ± 0.43 | 86.06 ± 0.32 | 70.94 ± 0.49 |
| **Meta-Learning** | MAML [4] | | 70.85 ± 0.38 | 56.59 ± 0.48 | 74.02 ± 0.41 | 59.17 ± 0.52 | 73.92 ± 0.36 | 58.02 ± 0.47 | 77.20 ± 0.38 | 61.40 ± 0.54 |
| | | *ft* | 73.84 ± 0.37 | 57.63 ± 0.47 | 80.19 ± 0.40 | 60.03 ± 0.51 | 78.82 ± 0.36 | 58.55 ± 0.48 | 84.74 ± 0.37 | 66.74 ± 0.52 |
| | | *cl* | 73.01 ± 0.36 | 56.69 ± 0.48 | 78.41 ± 0.40 | 61.16 ± 0.53 | 76.22 ± 0.35 | 58.32 ± 0.46 | 81.74 ± 0.38 | 63.61 ± 0.51 |
| | | *ft+cl* | 76.37 ± 0.37 | 59.36 ± 0.48 | 81.04 ± 0.39 | 63.88 ± 0.50 | 79.25 ± 0.34 | 62.84 ± 0.46 | 85.10 ± 0.39 | 67.70 ± 0.53 |
| | LEO [10] | | 74.49 ± 0.36 | 58.48 ± 0.48 | 80.25 ± 0.38 | 65.25 ± 0.51 | 75.86 ± 0.35 | 59.77 ± 0.47 | 82.15 ± 0.37 | 68.90 ± 0.49 |
| | | *ft* | 76.77 ± 0.35 | 60.52 ± 0.47 | 80.97 ± 0.36 | 65.44 ± 0.49 | 77.81 ± 0.34 | 61.81 ± 0.46 | 84.95 ± 0.36 | 69.59 ± 0.47 |
| | | *cl* | 74.66 ± 0.36 | 58.62 ± 0.46 | 80.74 ± 0.37 | 65.37 ± 0.50 | 76.13 ± 0.35 | 60.22 ± 0.47 | 82.31 ± 0.37 | 69.23 ± 0.48 |
| | | *ft+cl* | 71.91 ± 0.35 | 61.09 ± 0.47 | 81.43 ± 0.36 | 66.03 ± 0.48 | 77.72 ± 0.34 | 62.19 ± 0.45 | 85.04 ± 0.36 | 70.28 ± 0.47 |
| | MTL [12] | | 75.65 ± 0.35 | 58.49 ± 0.46 | 81.14 ± 0.36 | 64.29 ± 0.50 | 77.30 ± 0.34 | 62.99 ± 0.46 | 83.23 ± 0.37 | 70.08 ± 0.52 |
| | | *ft* | 77.17 ± 0.35 | 58.85 ± 0.44 | 82.01 ± 0.36 | 64.67 ± 0.47 | 79.40 ± 0.34 | 63.65 ± 0.45 | 84.76 ± 0.36 | 70.25 ± 0.49 |
| | | *cl* | 77.10 ± 0.34 | 58.86 ± 0.45 | 82.34 ± 0.36 | 66.70 ± 0.51 | 79.29 ± 0.35 | 63.14 ± 0.46 | 86.21 ± 0.37 | 70.16 ± 0.50 |
| | | *ft+cl* | 78.03 ± 0.33 | 61.17 ± 0.45 | 82.35 ± 0.35 | 65.72 ± 0.48 | 80.20 ± 0.33 | 64.40 ± 0.45 | 86.02 ± 0.35 | 71.45 ± 0.48 |
| | MN [13] | | 75.21 ± 0.35 | 61.05 ± 0.46 | 79.92 ± 0.37 | 66.01 ± 0.50 | 77.15 ± 0.36 | 63.45 ± 0.45 | 82.43 ± 0.37 | 70.38 ± 0.49 |
| | | *ft* | 75.52 ± 0.35 | 61.23 ± 0.45 | 80.18 ± 0.36 | 66.33 ± 0.49 | 77.80 ± 0.35 | 64.42 ± 0.46 | 83.82 ± 0.36 | 70.90 ± 0.50 |
| | | *cl* | 75.40 ± 0.34 | 61.14 ± 0.44 | 80.04 ± 0.35 | 66.26 ± 0.50 | 77.23 ± 0.35 | 64.21 ± 0.47 | 82.77 ± 0.35 | 70.61 ± 0.51 |
| | | *ft+cl* | 76.73 ± 0.34 | 62.64 ± 0.46 | 80.79 ± 0.35 | 67.30 ± 0.48 | 78.55 ± 0.36 | 64.89 ± 0.44 | 84.03 ± 0.36 | 71.41 ± 0.49 |
| | SIB [6] (transductive) | | 78.88 ± 0.35 | 67.10 ± 0.56 | 85.09 ± 0.35 | 77.64 ± 0.58 | 81.73 ± 0.34 | 71.31 ± 0.56 | 88.19 ± 0.34 | 81.97 ± 0.56 |
| | | *ft* | 79.58 ± 0.35 | 67.94 ± 0.55 | 85.12 ± 0.35 | 77.68 ± 0.57 | 82.00 ± 0.34 | 71.95 ± 0.56 | 88.20 ± 0.34 | 82.01 ± 0.56 |
| | | *cl* | 79.04 ± 0.33 | 67.77 ± 0.55 | 85.22 ± 0.35 | 77.72 ± 0.56 | 81.93 ± 0.35 | 71.66 ± 0.56 | 88.21 ± 0.33 | 82.01 ± 0.54 |
| | | *ft+cl* | 80.32 ± 0.35 | 68.85 ± 0.56 | 85.43 ± 0.35 | 78.03 ± 0.57 | 83.21 ± 0.33 | 73.51 ± 0.56 | 88.69 ± 0.33 | 83.07 ± 0.52 |
| | SIB [6] (inductive) | | 75.64 ± 0.36 | 57.20 ± 0.57 | 81.69 ± 0.34 | 65.51 ± 0.56 | 78.17 ± 0.35 | 60.12 ± 0.56 | 84.96 ± 0.36 | 69.20 ± 0.58 |
| | | *ft* | 76.23 ± 0.35 | 58.67 ± 0.56 | 82.04 ± 0.35 | 66.69 ± 0.57 | 79.34 ± 0.35 | 61.77 ± 0.56 | 85.24 ± 0.36 | 70.05 ± 0.57 |
| | | *cl* | 76.61 ± 0.35 | 58.12 ± 0.55 | 82.21 ± 0.35 | 66.28 ± 0.56 | 79.11 ± 0.35 | 61.25 ± 0.55 | 85.63 ± 0.34 | 69.90 ± 0.57 |
| | | *ft+cl* | 77.68 ± 0.34 | 60.33 ± 0.54 | 82.75 ± 0.35 | 67.34 ± 0.55 | 80.05 ± 0.34 | 63.14 ± 0.54 | 86.14 ± 0.34 | 71.45 ± 0.55 |

## A.6.2 Hardness-Specific Acc

(a) Linear

(b) Cosine

(c) *k*-NN

(d) MAML [4]

(e) LEO [10]

(f) MTL [12]

(g) Matching Net [13]

(h) SIB(transductive) [6]

(i) SIB(inductive) [6]

Figure A2: Supplementary to Figure 5. Hardness-specific Acc of 5-shot fine-tuning and meta-learning on *mini*ImageNet.

(a) Linear     (b) Cosine     (c) $k$-NN

(d) MAML [4]     (e) LEO [10]     (f) MTL [12]

(g) Matching Net [13]     (h) SIB(transductive) [6]     (i) SIB(inductive) [6]

Figure A3: Supplementary to Figure 5. Hardness-specific Acc of 5-shot fine-tuning and meta-learning on *tiered*ImageNet.

### A.6.3 CAM-Acc

Table A2: Supplementary to Figure 6. CAM-Acc (%) on fine-tuning and meta-learning. We used combined adjustment for IFSL.

| | Method | | ResNet-10 | | | | WRN-28-10 | | | |
| --- | --- | --- | --- | --- | --- | --- | --- | --- | --- | --- |
| | | | *mini*ImageNet | | *tiered*ImageNet | | *mini*ImageNet | | *tiered*ImageNet | |
| | | | 5-shot | 1-shot | 5-shot | 1-shot | 5-shot | 1-shot | 5-shot | 1-shot |
| Fine-Tuning | Linear | | $29.02 \pm 0.38$ | $25.22 \pm 0.38$ | $31.62 \pm 0.38$ | $31.05 \pm 0.39$ | $25.99 \pm 0.35$ | $24.74 \pm 0.34$ | $30.17 \pm 0.36$ | $29.76 \pm 0.37$ |
| | | +IFSL | $29.85 \pm 0.37$ | $26.67 \pm 0.38$ | $31.75 \pm 0.38$ | $31.43 \pm 0.37$ | $26.02 \pm 0.37$ | $24.96 \pm 0.36$ | $32.57 \pm 0.36$ | $30.64 \pm 0.38$ |
| | Cosine | | $28.10 \pm 0.37$ | $27.12 \pm 0.38$ | $29.82 \pm 0.37$ | $28.54 \pm 0.38$ | $27.54 \pm 0.38$ | $25.73 \pm 0.37$ | $32.60 \pm 0.35$ | $31.21 \pm 0.36$ |
| | | +IFSL | $28.18 \pm 0.37$ | $27.26 \pm 0.38$ | $31.38 \pm 0.37$ | $28.70 \pm 0.40$ | $27.82 \pm 0.38$ | $25.85 \pm 0.38$ | $33.65 \pm 0.37$ | $31.66 \pm 0.35$ |
| | $k$-NN | | $27.96 \pm 0.37$ | $26.65 \pm 0.37$ | $32.25 \pm 0.39$ | $30.36 \pm 0.39$ | $24.15 \pm 0.34$ | $23.30 \pm 0.33$ | $23.91 \pm 0.34$ | $21.99 \pm 0.33$ |
| | | +IFSL | $28.15 \pm 0.37$ | $26.81 \pm 0.37$ | $32.75 \pm 0.39$ | $30.84 \pm 0.39$ | $25.23 \pm 0.36$ | $24.14 \pm 0.35$ | $28.04 \pm 0.37$ | $26.46 \pm 0.37$ |
| Meta-Learning | MAML [4] | | $29.43 \pm 0.37$ | $27.39 \pm 0.38$ | $32.72 \pm 0.40$ | $32.14 \pm 0.40$ | $27.56 \pm 0.36$ | $26.46 \pm 0.36$ | $34.39 \pm 0.40$ | $31.07 \pm 0.39$ |
| | | +IFSL | $30.06 \pm 0.38$ | $28.42 \pm 0.38$ | $32.93 \pm 0.40$ | $32.24 \pm 0.39$ | $27.61 \pm 0.36$ | $26.91 \pm 0.38$ | $34.57 \pm 0.41$ | $31.22 \pm 0.40$ |
| | LEO [10] | | $30.24 \pm 0.38$ | $28.56 \pm 0.37$ | $31.64 \pm 0.38$ | $29.88 \pm 0.37$ | $29.15 \pm 0.38$ | $27.86 \pm 0.38$ | $31.27 \pm 0.37$ | $29.73 \pm 0.38$ |
| | | +IFSL | $30.67 \pm 0.37$ | $28.76 \pm 0.37$ | $32.01 \pm 0.38$ | $30.65 \pm 0.37$ | $29.20 \pm 0.37$ | $28.45 \pm 0.38$ | $31.98 \pm 0.39$ | $30.32 \pm 0.38$ |
| | MTL [12] | | $31.45 \pm 0.39$ | $30.13 \pm 0.39$ | $33.52 \pm 0.39$ | $33.11 \pm 0.39$ | $30.56 \pm 0.39$ | $29.78 \pm 0.40$ | $33.13 \pm 0.39$ | $32.35 \pm 0.39$ |
| | | +IFSL | $34.21 \pm 0.39$ | $31.59 \pm 0.40$ | $33.67 \pm 0.38$ | $33.50 \pm 0.39$ | $31.78 \pm 0.39$ | $30.12 \pm 0.39$ | $33.30 \pm 0.39$ | $32.64 \pm 0.39$ |
| | MN [13] | | $28.50 \pm 0.38$ | $28.42 \pm 0.39$ | $32.55 \pm 0.40$ | $31.88 \pm 0.39$ | $24.93 \pm 0.38$ | $25.34 \pm 0.39$ | $34.87 \pm 0.37$ | $29.10 \pm 0.38$ |
| | | +IFSL | $28.68 \pm 0.38$ | $28.77 \pm 0.38$ | $32.67 \pm 0.40$ | $32.10 \pm 0.40$ | $27.93 \pm 0.37$ | $25.81 \pm 0.37$ | $35.47 \pm 0.41$ | $30.71 \pm 0.39$ |
| | SIB [6] (transductive) | | $32.10 \pm 0.39$ | $31.19 \pm 0.39$ | $32.16 \pm 0.39$ | $30.49 \pm 0.39$ | $28.32 \pm 0.37$ | $26.76 \pm 0.38$ | $31.02 \pm 0.36$ | $28.43 \pm 0.38$ |
| | | +IFSL | $32.14 \pm 0.39$ | $31.34 \pm 0.39$ | $34.31 \pm 0.40$ | $32.59 \pm 0.40$ | $31.54 \pm 0.38$ | $29.82 \pm 0.36$ | $32.33 \pm 0.37$ | $30.26 \pm 0.39$ |
| | SIB [6] (inductive) | | $31.26 \pm 0.38$ | $30.56 \pm 0.39$ | $31.35 \pm 0.40$ | $30.48 \pm 0.39$ | $29.76 \pm 0.38$ | $28.02 \pm 0.37$ | $29.45 \pm 0.39$ | $27.98 \pm 0.39$ |
| | | +IFSL | $31.46 \pm 0.39$ | $30.78 \pm 0.40$ | $31.56 \pm 0.39$ | $30.89 \pm 0.40$ | $30.23 \pm 0.37$ | $28.75 \pm 0.39$ | $30.07 \pm 0.40$ | $28.57 \pm 0.39$ |

### A.6.4 Cross-Domain Evaluation

Table A3: Supplementary to Table 3. Acc (%) and 95% confidence interval averaged over 2000 5-way FSL tasks on cross-domain evaluation. Specifically, "*ft*" refers to feature-wise adjustment, "*cl*" refers to class-wise adjustment and "*ft+cl*" refers to combined adjustment.

| | Method | | ResNet-10 | | WRN-28-10 | |
|---|---|---|---|---|---|---|
| | | | 5-shot | 1-shot | 5-shot | 1-shot |
| **Fine-Tuning** | Linear | | $58.84 \pm 0.41$ | $42.25 \pm 0.42$ | $62.12 \pm 0.40$ | $42.89 \pm 0.41$ |
| | | *ft* | $60.12 \pm 0.39$ | $42.30 \pm 0.41$ | $63.13 \pm 0.39$ | $43.39 \pm 0.40$ |
| | | *cl* | $60.51 \pm 0.40$ | $42.43 \pm 0.42$ | $62.95 \pm 0.39$ | $44.21 \pm 0.40$ |
| | | *ft+cl* | $60.65 \pm 0.39$ | $45.14 \pm 0.40$ | $64.15 \pm 0.38$ | $45.64 \pm 0.39$ |
| | Cosine | | $58.30 \pm 0.39$ | $40.47 \pm 0.40$ | $60.21 \pm 0.39$ | $42.12 \pm 0.39$ |
| | | *ft* | $58.32 \pm 0.39$ | $41.01 \pm 0.40$ | $61.16 \pm 0.38$ | $42.35 \pm 0.41$ |
| | | *cl* | $58.68 \pm 0.39$ | $40.67 \pm 0.41$ | $61.87 \pm 0.40$ | $43.23 \pm 0.40$ |
| | | *ft+cl* | $60.23 \pm 0.38$ | $42.78 \pm 0.40$ | $62.49 \pm 0.38$ | $45.12 \pm 0.39$ |
| | $k$-NN | | $57.18 \pm 0.40$ | $38.44 \pm 0.37$ | $59.31 \pm 0.41$ | $40.53 \pm 0.42$ |
| | | *ft* | $59.44 \pm 0.39$ | $43.49 \pm 0.40$ | $62.48 \pm 0.39$ | $45.68 \pm 0.43$ |
| | | *cl* | $58.37 \pm 0.39$ | $43.20 \pm 0.41$ | $62.04 \pm 0.39$ | $45.36 \pm 0.40$ |
| | | *ft+cl* | $59.59 \pm 0.40$ | $43.45 \pm 0.40$ | $62.45 \pm 0.40$ | $45.72 \pm 0.40$ |
| **Meta-Learning** | MAML [4] | | $51.09 \pm 0.43$ | $37.20 \pm 0.46$ | $55.04 \pm 0.42$ | $39.06 \pm 0.47$ |
| | | *ft* | $54.95 \pm 0.44$ | $37.34 \pm 0.47$ | $59.57 \pm 0.44$ | $39.25 \pm 0.46$ |
| | | *cl* | $53.62 \pm 0.43$ | $38.13 \pm 0.47$ | $56.80 \pm 0.45$ | $40.32 \pm 0.48$ |
| | | *ft+cl* | $56.71 \pm 0.46$ | $40.36 \pm 0.46$ | $60.89 \pm 0.45$ | $42.16 \pm 0.47$ |
| | LEO [10] | | $56.52 \pm 0.46$ | $39.21 \pm 0.53$ | $56.66 \pm 0.48$ | $41.45 \pm 0.54$ |
| | | *ft* | $56.77 \pm 0.48$ | $39.72 \pm 0.54$ | $62.95 \pm 0.47$ | $45.46 \pm 0.55$ |
| | | *cl* | $56.73 \pm 0.47$ | $40.12 \pm 0.55$ | $56.90 \pm 0.47$ | $41.93 \pm 0.56$ |
| | | *ft+cl* | $61.27 \pm 0.46$ | $42.79 \pm 0.52$ | $63.30 \pm 0.47$ | $43.81 \pm 0.56$ |
| | MTL [12] | | $56.61 \pm 0.42$ | $41.56 \pm 0.43$ | $56.89 \pm 0.41$ | $43.15 \pm 0.44$ |
| | | *ft* | $61.34 \pm 0.41$ | $42.90 \pm 0.43$ | $63.49 \pm 0.40$ | $45.28 \pm 0.44$ |
| | | *cl* | $60.62 \pm 0.41$ | $42.87 \pm 0.42$ | $62.94 \pm 0.40$ | $45.57 \pm 0.43$ |
| | | *ft+cl* | $62.39 \pm 0.40$ | $44.51 \pm 0.43$ | $65.00 \pm 0.40$ | $46.67 \pm 0.43$ |
| | MN [13] | | $53.39 \pm 0.46$ | $40.34 \pm 0.56$ | $53.08 \pm 0.45$ | $42.04 \pm 0.57$ |
| | | *ft* | $54.22 \pm 0.46$ | $40.62 \pm 0.57$ | $54.97 \pm 0.47$ | $42.52 \pm 0.58$ |
| | | *cl* | $53.72 \pm 0.47$ | $40.42 \pm 0.56$ | $53.43 \pm 0.45$ | $42.19 \pm 0.56$ |
| | | *ft+cl* | $56.03 \pm 0.45$ | $41.68 \pm 0.54$ | $58.69 \pm 0.44$ | $43.58 \pm 0.56$ |
| | SIB [6] (transductive) | | $60.60 \pm 0.46$ | $45.87 \pm 0.55$ | $62.60 \pm 0.49$ | $49.16 \pm 0.58$ |
| | | *ft* | $61.12 \pm 0.45$ | $46.64 \pm 0.55$ | $63.15 \pm 0.47$ | $49.78 \pm 0.56$ |
| | | *cl* | $60.70 \pm 0.46$ | $46.14 \pm 0.56$ | $63.02 \pm 0.48$ | $49.43 \pm 0.57$ |
| | | *ft+cl* | $62.07 \pm 0.44$ | $47.07 \pm 0.53$ | $64.07 \pm 0.49$ | $50.71 \pm 0.54$ |
| | SIB [6] (inductive) | | $59.06 \pm 0.42$ | $41.48 \pm 0.43$ | $59.94 \pm 0.42$ | $43.27 \pm 0.44$ |
| | | *ft* | $59.45 \pm 0.41$ | $41.98 \pm 0.44$ | $60.33 \pm 0.44$ | $43.61 \pm 0.45$ |
| | | *cl* | $59.32 \pm 0.42$ | $41.67 \pm 0.43$ | $60.46 \pm 0.43$ | $43.52 \pm 0.45$ |
| | | *ft+cl* | $59.89 \pm 0.41$ | $43.20 \pm 0.43$ | $61.45 \pm 0.43$ | $44.27 \pm 0.44$ |

## Footnotes

[1]https://github.com/wyharveychen/CloserLookFewShot

[2]https://github.com/yaoyao-liu/meta-transfer-learning

[3]https://github.com/deepmind/leo

[4]https://github.com/hushell/sib_meta_learn