[Reviews · NeurIPS 2020]

Review 1

Summary and Contributions: The authors first reveal that the pre-trained knowledge could be confounders that cause spurious feature-label correlations and thus negative transfer. To address this, the authors propose to intervene in the feature transfer of few-shot learning through backdoor adjustment based on the SCM model. The proposed method is shown to improve the existing fine-tuning and meta-learning based FSL methods.

Strengths: The paper is well-structured and easy to follow. The addressed issue of feature-label confounder is important in few-shot learning. 1) The authors revealed the negative transfer in few-shot learning caused by spurious feature-label correlation. 2) A novel causality-based few-shot learning approach is proposed to intervene in the feature transfer and to alleviate the spurious correlations through backdoor adjustment using SCM model. 3) The authors experimentally show the proposed method benefits the existing few-shot methods and prevents negative transfer.

Weaknesses: 1. The authors blame the performace drop caused by the spurious correlations to the stronger pre-trained feature extractor, which seems unreasonable. It's the classifier fails to get rid of the spurious correlation, not the feature extractor. 2. N equal-sized disjont subsets are divided. How to guarantee that the causal factors will not coexist cross subsets? 3. How to learn the feature selector? 4. More convincing experiments. 1) Lacking ablation study to show how the Feature-wise and Class-wise Adjustment individually would influence the performance; 2) Better to compare to other causality-based strategies that reduce such spurious feature-label correlation, e.g., under the potential outcome framework.

Correctness: Yes

Clarity: Yes

Relation to Prior Work: Yes

Reproducibility: No

Additional Feedback: The assumptions on the n equal-size disjoint subsets need to be clarified and the experimental results need to be more convincing. This is an interesting paper, and my concerns are mostly addressed in the rebuttal.


Review 2

Summary and Contributions: The paper address the few-shot learning problem setup from a causal-inference perspective. The paper main contribution is proposing a causal graph that describes the few-shot learning process. The graph suggests that image classification according to p(class|image) may be confounded by the pre-training "knowledge". Therefore, the approach proposes to classify an image according to p[class | do(image) ] using the backdoor adjustment formula. Then it suggests to use a mapping that approximates the adjustment formula to make few-shot classification. Overall, the ideas are very interesting and definitely worth pursing. However many times the paper assumptions are not well defined nor explained or justified, and I found it hard to validate the correctness of main parts of the work.

Strengths: +Proposing to model the causal-graph of few-shot learning process is novel. +The reported improvement over SoTA of FSL benchmarks is significant.

Weaknesses: --Clarity is the major issue of this paper. See more details under "Clarity" --Due to the clarity issues I couldn't verify the correctness of main parts of the work. --Prior work on causal relations between images and labels is not discussed nor cited, more details under "Relation to prior work".

Correctness: Due to clarity issues I couldn't verify the correctness of main parts of the work. See my response to "Clarity". Other correctness issues: --Line 146 and Figure 2.b: If all the lion images of a *test* support set would be pictured with a grass background, the grass feature may still be used as a predictive signal for classification, do(X) would not make it disappear, regardless of the number of image samples used for training. Figure 2.a: Not sure the degradation with "strong Omega" is significant. (1) Missing errors bars for S!~Q. (2) How did the authors identify the dissimilar query sets?

Clarity: My most major concern is about the SCM in section 2.2: The proposed causal graph is not well defined. The nodes in the graph should correspond to well-defined random variables and each edge should be justified. This is not the case in the paper: D is loosely defined to be "the dataset and its induced model". What a sample of D stands for? Is it a dataset from a corpus of datasets? is it a pretrained model weights? is it both? X is loosely defined to be the "feature representation". What a sample of X stands for? Is it all the activations of some neural-net? It would be beneficial to state how pixels are mapped to X , and then explain how X is it related to the support and query set. Are the support and query set sampled from D (since D cause X)? are they related to I? How does the test support set is used? Does it replaces D? or is it related to I? What a sample of C stands for? In what it differs from X? If C is a transformed representation of X, then how come there is a direct edge of D cause C? What is the output of g(x,d)? is it a set of indices as defined (Line 193)? A vector of features (Line 207)? Which edges do the learned weights affect? It would be beneficial to write the structured equation model of the graph, and give a concrete example with a well-known FSL model, of how each one of the mapping in the SCM is related to the training process of that FSL model. Section 2.3: --Why there is a backward edge X-->I ? --Why MSL is different than FSL? (Why increasing the number of samples changes the causal graph?) Other clarity issues: (they may be resolved if above issues are well explained) --Why there is an edge X->Y? isn't it sufficient to have X->C->Y? --Line 192: would be beneficial to explicitly write what [x]_c means. i.e. [x]_c = … 1/<number of features>? Shouldn't it be a prior over the some distribution of the data? --Lines 83-92: It is worth to explicitly define what theta is used for and what phi is used for. --line 86: what is the difference between P(y|x;phi) to P_{phi}(y|x;theta) in line 87?

Relation to Prior Work: Prior work on modelling the causal relations between images and labels is not discussed nor cited. Most of prior work [1-4] argue that such a setup is "anti-causal" (labels cause images), while [5] argues (in a side note) that this setup may be causal (images cause labels). Since this work takes debatable the side of [5] I suggest the paper should discuss and better justify that perspective. Other uncited works already raised the difficulties with recognition due to spurious correlations, e.g. [6] [1] Heinze-Deml, Conditional Variance Penalties and Domain Shift Robustness [2] Gong, Domain adaptation with conditional transferable components. [3] Sholkopf, On Causal and Anticausal Learning [4] Kilbertus, Generalization in anti-causal learning [5] Arjovsky, Invariant Risk Minimization [6] Beery, Recognition in Terra Incognita

Reproducibility: No

Additional Feedback: Typo Fig 3. "ashcan" --> "trashcan"? POST REBUTTAL COMMENTS: I thank the authors for their response. Although I think causal ideas for FSL are very interesting and definitely worth pursing. I still think that this work is not ready for publication in its current state. I still find that the analysis and the approach have major clarity issues, which strongly hinder the evaluation of correctness. In addition, I didn't find the reply about the claims in Fig 2a & Fig2b to be satisfactory. I will stress that it is still unclear/missing: 1. *Why* the edges in the suggested causal graph are reasonable. 2. How do they reflect the causal mechanism in FSL? 3. What type of confounding effects can the approach resolve, and *why* (this point is very important) . 3.a. As I explained in the review, I was not convinced that the approach can resolve the type of confounder in the example in Fig2.b, because that confounding effect is not related to the number of seen images (MSL vs FSL). 4. Having "a concrete example with a well-known FSL model, of how each one of the mapping in the SCM is related to the training process of that FSL model." There are still other major clarity issues, and I don't think that the model proposed by this paper can be reproduced. For example, * If d_i is a subset of feature channels, how this subset is sampled from the set of all feature channels? * I didn't get a clear answer what is the difference between X, C. Why would they manifest differently given a specific d_i. * If I->X then do(X) neglects the edge I->X. Therefore p[Y|do(X)] would ignore a specific image. * Fig 2a (Rebuttal L15) - Error bars - the authors didn't reply to my question about statistical significance of their claims in Fig 2.a * The meaning of the random variables (D,X,C) is still unclear (e.g. on line 193, C=g(x,d) is a set of indices, while on line 207 it is a vector of features)


Review 3

Summary and Contributions: This paper points out a systematic deficiency caused by pre-trained knowledge overlooked by existing Few-Shot Learning works and explains the problem with a Structural Causal Model (SCM) based on the causalities among the pre-trained knowledge, few-shot samples, and class labels. The authors propose a novel paradigm named Interventional Few-Shot Learning (IFSL), which is agnostic to the existing FSL methods. IFSL improves all baseline methods by a considerable margin on three widely-used FSL benchmarks.

Strengths: The authors proposed an excellent and novel cut by considering the possible defects caused by the pre-trained knowledge on different backbones and similarities between Support set (S) and Query set (S). This work provides reasonable explanations and explorations for the systematic deficiency caused by pre-trained knowledge with a Structural Causal Model (SCM) through analyzing the causalities among the pre-trained knowledge, few-shot samples, and class labels. IFSL brings a considerable improvement, which is orthogonal to existing fine-tuning or meta-learning based FSL methods, and the experiments have validated that IFSL can be used with many kinds of FSL methods, which achieves a new state-of-the-art on 1/5-shot learning by improving all baselines.

Weaknesses: As 4-Conv-Block backbones are commonly used during the early works of FSL, such as MAML, Matching Networks, and Prototypical Networks, it would be more robust to compare the 4-Conv-Block backbone besides ResNet-10 and WRN-28-10 to indicate what would be led by the systematic deficiency with a weaker backbone. Fine-Tuning and Meta-Learning both suffer the systematic deficiency but may differ in extent. IMHO, the meta-learning suffers less, as it reduces the effect of the pre-trained knowledge through the meta-learning phase with a large number of training episodes and pays more attention to how to learn new tasks. More intuitions or explanations on the differences of the bad side of pre-trained knowledge in the two paradigms would be very helpful.

Correctness: I have not gone through all details in the appendix but found no flaw by following the authors' descriptions in the main paper.

Clarity: I really enjoyed reading this manuscript. A minor typo: Line 260: ``3)'' is missing before SIB.

Relation to Prior Work: Related work has been discussed moderately. It would be interesting to see empirical comparisons to present works that aim at the negative transfer problem.

Reproducibility: Yes

Additional Feedback: Which FSL paradigm Fig. 2 is targeting at, Fine-Tuning or Meta-Learning? ==== Thank the authors for their rebuttal. I keep my original score and encourage the authors to adequately address those questions in other reviews.


Review 4

Summary and Contributions: This paper explores the relationship between pre-training and few-shot learning. Prior approaches to few-shot learning have a pre-training approach followed by an adaptation stage. The authors point out a deficiency in this methodology. Features learned in the pre-training can mislead the few-shot fine-tuning/adaptation stage, especially when the query is dissimilar to the support set instances in the few-shot task. The authors first present a causal model to explain this behavior. Second, a practical implementation based on this model is proposed to alleviate the issue in few-shot learning methods called Interventional Few-shot Learning (IFSL). IFSL is shown to improve over many popular few-shot learning approaches, across all regimes in similarity/dissimilarity between query and support set. Evidence is provided in the form of visualization which suggests that IFSL helps few-shot learners focus on the right features to make predictions as opposed to spurious features introduced by pre-training.

Strengths: * Makes an important contribution in the few-shot learning space. * Strong theoretical basis to back claims. * A general approach is proposed based on the theoretical framework which helps improve many popular few-shot learning methods. * Strong experiments that showcase the advantages of the proposed method.

Weaknesses: * Paper is not easily accessible to the reader not familiar with causal modeling/inference. Introducing some preliminaries could help understand some key principles (Eg. backdoor adjustment) and enable the ideas to reach a wider audience.

Correctness: Claims and experiments are sound to my knowledge. Although there were some parts I did not fully understand.

Clarity: Paper is generally well written.

Relation to Prior Work: Contributions of the paper and relevance to prior work are clearly discussed. References are adequate.

Reproducibility: Yes

Additional Feedback: Very interesting paper, I enjoyed reading it. Work is well motivated, an elegant solution is proposed and the method is backed by strong empirical evidence. I felt like some of the design choices for feature-wise adjustment and class-wise adjustment in section 3 could be motivated/justified better (Eg. choice of g(x, d)). Table 2 caption is not informative enough, and the purpose of Table 2 is not clear enough from line 291 alone.

[Author Response · NeurIPS 2020]

We thank all reviewers for finding our paper *novel*, *interesting*, and with *strong* performance (**R1**, **R2**, **R3**, **R4**). We
apologize for missing the details of causal graph and some related references (**R2**). We will address all the concerns.
**R1-Q1** **Cause of Performance Drop**. You are correct. We will rephrase to highlight the blame is on classifier.
**R1-Q2** **Coexistence of Causal Factors**. In fact, the underlying assumption of using feature channels is that they are
*Independent Mechanisms* (IM) [A], which generate $X$ and $Y$ (see details in **R2-Q6**) and there could be confounders
across the subsets of channels. Fortunately, those confounders have no direct causal links to $X$ and $Y$ [B] and thus
adjusting the channels can block the effect from the confounders (Markov factorization).
**R1-Q3** **Feature Selector**. Feature selector $c$ is a pre-defined mask for selecting feature channels (see line 193-195).
**R1-Q4** **More Convincing Experiments**. Actually, we have provided the results in section A.6 of the supplementary
material, where feature-wise and class-wise adjustment gain similar improvements on average.
**R1-Q5** **Other Causality-Based Strategies**. The analysis in our paper can include Rubin's potential outcome framework,
*e.g.*, using propensity score as another deconfounding approach, besides class-/feature-wise adjustment.
**R2-Q1** **Figure 2b**. We want to clarify that in the backdoor adjustment, $do(X)$ does not make the "grass" feature
disappear, *i.e.*, it is still used as a predictive signal but with its contribution adjusted by $P(\text{"grass"})$.
**R2-Q2** **Figure 2a**. We will add the error bars. The dissimilarity is measured by query hardness defined in line 265.
**R2-Q3** **Formal Definition of Causal Graph**. Sorry for the clarity issue. We omitted this as we intended to only
offer a high-level concept for readers with CV/ML background. We will follow your suggestion to provide a formal
and well-defined SCM in revision. Specifically, we model FSL as a Structural Causal Model $\mathcal{M}$ that consists of a
collection $\mathcal{M} = (f_X, f_C, f_Y)$ of structural assignments $X := f_X(I, D), C := f_C(X, D), Y := f_Y(X, C)$. $D$ is
defined as the stratum set of pre-trained knowledge $D = \{d_1, \ldots, d_n\}$ learnt from large dataset $\mathcal{D}$, where $d_i$ is a subset
of feature channels in feature-wise adjustment (*FT*) or a pre-training class in class-wise adjustment (*CL*). The sample ID
$I = \{1, \ldots, |\mathcal{S}|\}$ in training and $I = \{1, \ldots, |\mathcal{Q}|\}$ in testing, where $\mathcal{S}$ is support set and $\mathcal{Q}$ is query set ($\mathcal{S}, \mathcal{Q} \cap \mathcal{D} = \emptyset$).
$f_X$ uses deep network to obtain feature $X$ for the image with ID $I$. $f_C$ projects $X$ on a stratum of knowledge $D = d_i$
to get the image-specific $C$ representation (see line 193, 207). The classification logits $Y$ is given by $f_Y$. We are
sorry for the confusion on $X \to Y$ and we will highlight in revision that $X \to C \to Y$ is sufficient in *FT*, while in
*CL*, $X \to Y$ is necessary as the class-based $C$ might be an incomplete representation of $X$. The objective of FSL
is $P(Y|do(X))$ and the parameters of $f_Y$ is learnt in training. Our model is generally applicable to fine-tuning and
meta-learning, where they differ in the parameterization of $f_Y$: $\theta$ in fine-tuning (see line 79) and an *additional* set of
parameters $\phi$ in meta-learning (see line 83).
**R2-Q4** **Explicit Form of** $[\mathbf{x}]_c$. $[\mathbf{x}]_c = \{x_i\}_{i \in c}$, where $x_i$ is the value of feature vector $\mathbf{x}$ at $i$-th position.
**R2-Q5** **Backward Edge** $X \to I$. For example, in the 1-shot extreme case of FSL, there is a 1-to-1 mapping between
sample ID $I$ and feature $X$, denoted as the bi-directed edge $I \leftrightarrow X$ in Figure 4(b). However, in MSL where training
data is abundant, $X \to I$ is cut off because tracing the ID given feature $X$ is practically impossible (see line 148).
**R2-Q6** **Causal or Anti-Causal**. We will discuss your suggested related work as follows, *i.e.*, why do we adopt $X \to Y$
(causal) not $Y \to X$ (anti-causal) in FSL? Anti-causal learning [C] is based on the Independent Mechanisms (IM)
or causal generative factors assumption [A][B], which states that the observations are generated from IM. Therefore,
when label $Y$ is simply disentangled enough to be IM (*e.g.*, 10 digits in MNIST [C]), $Y \to X$ establishes. However, in
our FSL, when the label is much more complex, *e.g.*, the ImageNet labels "dog" and "cat" are semantically entangled
such as "soft fur", we should consider the causal prediction $X \to Y$ as it is essentially a reasoning process , *e.g.*, there
are recent empirical justifications of $X \to Y$ in complex CV tasks [D]). In this way, the IM becomes the $D$ in our
method, where $D$ generates visual features $X$ and $D \to Y$ emulates our human's naming process, *e.g.*, using "small,
fur, four-legged" to name "meerkat". Note that, although each piece of knowledge in $D$ is also complex, CNN has
"engineered" them to be disentangled, such as the feature channels (feature-wise adj.) and softmax class responses
(class-wise adj.). We will also explore the combination of anti-causal and causal predictions in future work, *e.g.*,
following [E] when $Y$ is not perfectly disentangled or entangled.
**R3-Q1** **Backbone Choice**. Thanks, we will validate on weaker backbones following your suggestion.
**R3-Q2** **Why meta-learning suffers less?** We totally agree with your opinion that meta-learning suffers less from the
deficiency and actually we validated this intuition in our experiments (see line 283, 299). We have also discussed the
potential reason that meta-learning is essentially a form of intervention (see line 284).
**R3-Q3** **Negative Transfer**. We will revise and add empirical comparisons to negative transfer literature.
**R3-Q4** **Figure 2**. Sorry for the confusion. Figure 2 targets fine-tuning and we will revise to highlight this.
**R4-Q1** **Preliminaries**. Thanks for the suggestion. We will provide a more detailed introduction to preliminaries during
revision, such as the formal definition of the casual graph in **R2-Q3**.
**R4-Q2** **Design Choices**. The design choices for feature-/class-wise adjustment reflect the motivation discussed in line
179-185: *e.g.*, in class-wise adjustment, $g(\mathbf{x}, d)$ is the distilled pre-trained knowledge.
**R4-Q3** **Table 2 Clarity**. Sorry for the clarity issue. We will revise the caption of Table 2 to highlight its purpose.

[A] Parascandolo et al. *Learning independent causal mechanisms.* ICML'18 [B] Suter et al. *Robustly Disentangled Causal Mech-*
*anisms: Validating Deep Representations for Interventional Robustness.* ICML'19 [C] Heinze-Deml et al. *Conditional Variance*
*Penalties and Domain Shift Robustness.* arXiv [D] Qi et al. *Two causal principles for improving visual dialog.* CVPR'20 [E] Sid-
dharth et al. *Learning disentangled representations with semi-supervised deep generative models.* NeurIPS'17


[Meta-Review · NeurIPS 2020]

This paper analyzes few-shot learning from a causal inference perspective and presents an interesting claim that pretrained knowledge is a confounder that limits performance. The authors use this finding to propose an interventional few shot learning paradigm. I think this is a solid paper where the theoretical insight results in a good empirical performance. R2 has a serious concern regarding the clarity of the paper that makes it difficult to verify the correctness of the paper, in particular with respect to the kind of confounding effect that this approach can resolve. I think this is a valid concern and I suggest the authors attempt to fully address this in the final version of the paper.